

The effect of low density over the "roof of the world" Tibetan Plateau on the
triggering of convection
Yinjun Wang[1], Xiangde Xu[1], Mingyu Zhou[2], Donald H. Lenschow[3], Xueliang Guo[1],
Yang Zhao[1], and Bin Chen[1]
[1] State Key Laboratory of Severe Weather, Chinese Academy of Meteorological
Sciences, Beijing, China,
[2] National Marine Environmental Forecasting Center, Beijing, China,
[3] National Center for Atmospheric Research, Boulder CO, USA
Correspondence to: Xiangde Xu (xuxd@cma.gov.cn)
Abstract
We study the relationships between convective characteristics and air density over the
Tibetan Plateau (TP) from the perspective of both climate statistics and large eddy
simulation (LES). First, based on climate data, we found that there is stronger thermal
turbulence and higher frequency of low cloud formation for the same surface relative
humidity over the eastern and central TP compared with the eastern monsoon region
of China. Second, we focus on the dynamical and thermal structure of the atmospheric
boundary layer (ABL) with low air density. With the same surface heat flux, a
decrease in air density enhances the buoyancy flux, which increases the ABL depth
and moisture transport from the subcloud layer into the cloud layer. With the same
low cloud cover for different air densities, the greater ABL depth for lower air density
means that the average mixed-layer relative humidity with higher air density will be
greater than that with low air density. Results from a subcloud convective velocity
scaling scheme were compared with LES results, which indicated that the original
fixed parameter values in this scheme may not adequate in case of lower relative
humidity and weaker thermal turbulence in the subcloud layer.

Key words: Tibetan Plateau, air density, convective boundary layer, shallow cumulus,
large eddy simulation

1 Introduction
The Tibetan Plateau (TP), which resembles a "third pole" and a "world water
tower", plays an important and special role in the global climate and energy–water
cycle (Xu et al., 2008). Cumulus convection over the TP transfers heat, moisture and
momentum to the free troposphere, which can impact the atmospheric circulation
regionally and globally (Li and Zhang 2016). (Dai,1990) conducts statistics of the



proportion of different cloud types in different regions, the results show that cumulonimbus clouds over the center of the TP account for 21%, which is about five times than that of over the rest of China. The elevated land surface with strong radiative heating makes the massive TP a favorable region for initiating numerous convective cells, and has a high frequency of cumulonimbus or mesoscale convective systems (MCSs) (Sugimoto and Ueno, 2012). Li and Zhang (2016) confirmed that the climatological occurrence of cumulus over the TP is significantly greater than over the surrounding area by using four years of CloudSat–CALIPSO satellite data. They found that the ubiquitous cumulus over the northern TP is related to the higher air temperature and larger relative humidity above the surface than those in the surrounding regions at the same height above sea level.

Xu et al. (2002) and Zhou et al. (2000) found that the turbulence with motion at vertical speeds of up to 1 m s$^{-1}$ at a height of about 120 m above the surface with the horizontal scale about 600 m, strong convective plumes and a larger than 2000 m mixed layer may result in ubiquitous "pop-corn-like" convective clouds over the TP. These clouds, which have relatively large vertical and small horizontal scale occurred when the strong vertical motions penetrate the capping inversion layer. They can sometimes evolve into mature super convective cloud clusters. Xu et al. (2002) documented the structure of the distinctive "popcorn-like" cloud systems over the TP by comprehensively analyzing TIPEX II observational data. Xu et al. (2012) conjectured that these clouds may be favored by low air density $\rho$ and strong turbulence. The reduced $\rho$ and enhanced buoyancy production results in turbulent characteristics of the convective boundary layer (CBL) over the plateau that are considerably different from that over the plain (Xu et al., 2012).

The sodar data from TIPEX II and the boundary layer tower data from TIPEX III indicated that the contributions of buoyancy and shear to the turbulence kinetic energy in the lower troposphere were larger over the TP than over the southeastern margin of the TP and the low-altitude Chengdu Plain (Zhou et al., 2000; Wang et al., 2016). Observations also indicated that organized turbulence on the meso- and micro-scale was large due to the abnormally strong solar radiation over the TP (Zhou et al., 2000). Therefore, the question arises as to whether there is a relationship among the formation and evolution of frequent "pop-corn-like" convective clouds, low $\rho$, and turbulence generation over the TP.

We discuss the above key scientific issues from two aspects: climate statistics and large eddy simulation (LES). Climate statistics are used to study the spatial distributions of summertime low cloud cover (*LCC*). Here low cloud includes the total




area of observed cloud cover with cloud base less than 2.5 km above the ground level.
In the early stages of development we classify the "pop-corn-like" convective cloud as
shallow convective cumulus due to their very small horizontal scale (from tens of
meters to a few kilometers). We also use high-resolution LES to simulate the
three-dimensional turbulent flow and shallow cumulus. We conducted sensitivity tests
to study the effect of varying $\rho$ on the formation and evolution of shallow cumulus,
and the interrelation between turbulence and convective motion. The LES is also used
to test subcloud convective velocity scaling schemes with varying $\rho$. Finally, we
attempt to explain some of the physical processes determining the climate statistics of
summertime low cloud cover over China.

2  Data
The data used here are taken from the following observation and reanalysis products:
2.1 observational data at Nagqu in TIPEX III
a)  Turbulent fluxes (sensible and latent heat flux) in the surface layer calculated

by the eddy covariance technique. Here we applied EDDYPRO software for

eddy covariance data quality control. EDDYPRO is an open source software

application developed, maintained and supported by LI-COR Biosciences

(Available at www.licor.com/EddyPro).

b)  L-band sounding data from the China Meteorological Administration (CMA)

operational station, three times a day at 06:00, 12:00 and 18:00 LST.

c)  Daily mean climate data from 2479 automatic weather stations (AWS) from

1979 to 2016 in China.

2.2 ERA-Interim reanalysis data
We used the synoptic monthly means derived from the ERA-interim reanalysis
surface-layer data every 3 hours for summer from 1979 to 2016. We also used 9 day
ERA-interim reanalysis data at standard isobaric levels every 6 hours in summer 2015
to calculate large-scale forcing for the LES. Both of these data sets have a spatial
resolution of 0.75 ° x 0.75 °, and all the final results of large-scale forcing are derived
from the mean values of grids within a radius of 300 km.

3  The climatic characteristics of summertime low cloud and their correlation with
air density
Figure 1(a) presents the following two consistent patterns: 1. Because the atmosphere
is easily moistened to saturation with ambient high relative humidity, the $LCC$
generally increases with increasing relative humidity at 2 m ($RH_{2m}$) with constant air



density at 2 m ($\rho_{2m}$), which is consistent with our common sense. 2. The $LCC$
increases approximately parabolically with decreasing $\rho_{2m}$ for $RH_{2m}$ both greater than
and less than 75% (corresponding to A region and B region, respectively). That is,
more low cloud exists in the high altitude area of the TP with low $\rho_{2m}$ and low $RH_{2m}$
($50\% < RH_{2m} < 70\%$). As shown in Figure 1(b), $LCC$ greater than 50% is mainly over
mid-eastern TP, and southwestern China. Despite the abundant water vapor over the
eastern China monsoon region (ECMR), $LCC$ is significantly lower over the ECMR
relative to that over the mid-eastern TP. Figure 1(c) shows that there is a large area of
$LCC$ greater than 35% north of 30 ˚N in mid-eastern TP with low $RH_{2m}$ ($RH_{2m} < 70\%$)
and large surface virtual potential temperature flux $\overline{\left(w'\theta_v'\right)_s}$
$> 0.1$ ˚K m s⁻¹ in contrast to the low altitude area.

4   The effect of low air density over the TP on the formation and development of
cumuli
Zhu et al. (2002) indicated that shallow cumuli result from the daytime
development of the CBL in which buoyancy is the dominant mechanism driving
turbulent mixing. Mixing by convective elements, or thermals, is limited by the height
of the mixed layer $h$, which is capped by an overlying inversion layer. Whether or not
shallow cumuli can form is determined by the thermodynamic properties of the CBL
and maximum height of the convective turbulence. In Appendix A we present a simple
dry CBL model that illustrates the sensitivity of $h$ to air density at the surface. We
explore this sensitivity in more detail for the TP region by using LES to analyze the
effect of $\rho$ on the formation and evolution of cumuli. The LES model description and
simulation setup, and a comparison between the observations and the LES are shown
in Appendix B. We analyze in detail the results of control experiment CON, six
sensitivity experiments with air densities ($1.2\rho_{CON}$, $1.4\rho_{CON}$ , $1.7\rho_{CON}$) and relative
humidities ($1.4\rho_{CON}RH0.05$, $1.4\rho_{CON}RH0.15$, $1.4\rho_{CON}RH0.3$).
4.1 The height of the mixed layer $h$ and its growth rate $dh/dt$ for varying air density
Zhu et al (2002) showed that with constant water vapor mixture ratio $q_T$ and
adiabatic temperature lapse rate $\partial T/\partial z = -\gamma_d$ within the CBL, the relationship between
the relative humidity at the top of the surface layer $RH_0$  and the relative humidity at
the top of the mixed layer (ML) $RH_h$ can be written as

$$RH_h \approx RH_0 \left( 1 + \frac{L\gamma_d h}{R_v T_0^2} \right)$$


(1)

where $L$ is the enthalpy of vaporization, $R_v$ is the gas constant of water vapor, and $T_0$ is





the temperature at the top of the surface layer. Eq. (1) indicates that $RH_h$ increases
with increasing $h$ under conditions of fixed $T_0$ and $RH_0$. In this study we use the
profiles of virtual potential temperature gradient $\partial\theta_v/\partial z$ to define $h$ as the lowest level
for which $\partial\theta_v/\partial z > 2$ K km$^{-1}$.
The equation for the rate of change of $h$ is given by Betts (1973) and Neggers et
al. (2006) as

$$\frac{dh}{dt} = w_e + w_s - M \tag{2}$$

where $w_e$ and $w_s$ are the entrainment and large scale subsidence velocities,
respectively and $M$ is the kinematic mass flux of air transported by clouds from the
subcloud to cloud layer. $M$ can be modeled as

$$M = a_{cc} w_{cc} \tag{3}$$

where $a_{cc}$ and $w_{cc}$ are the maximum cloud core fraction and its corresponding vertical
velocity at the same height, respectively. Cloud core is the positively buoyant region
with respect to the environment. Here we ignore the differences between the height of
maximum $a_{cc}$ and $h$ when we use eq. (2) to calculate $w_e$. $M$ can be ignored since it is
more than an order of magnitude smaller than $w_e$ when $a_{cc} < 1\%$(before about 15:00
LST). However, $M$ cannot be ignored in the developmental stage of cumuli due to
larger $a_{cc}$ (after about 15:00 LST), which will be discussed in the subsequent section
4.3. $w_s$ is significantly smaller than $w_e$ in this study, and thus the variation of $dh/dt$
mainly depends on $w_e$. Figure 2(b) shows the time variations of $w_e$ calculated with eq.
(2) in four LES experiments (CON, $1.2\rho_{CON}$, $1.4\rho_{CON}$, $1.7\rho_{CON}$). For the zero-order
jump assumption, $w_e$ can be modeled as:

$$w_e = -\frac{\overline{(w'\theta_v')}_h}{\Delta_{\theta_v}} = \frac{\beta_1 \overline{(w'\theta_v')}_s}{\Delta_{\theta_v}} \tag{4}$$

where $\overline{(w'\theta_v')}_h$ is the entrainment flux at the top of the CBL, $\overline{(w'\theta_v')}_s$ is the surface
buoyancy flux, $\Delta_{\theta_v}$ is the virtual potential temperature difference across the inversion,
and the proportionality factor $\beta_1$ is assumed to be a constant, ~0.2 for free convection
(e.g. Sullivan et al., 1998). For constant $\beta_1$ Zhu et al (2002) derived the following
expression for $\Delta_{\theta_v}$:

$$\Delta_{\theta_v} = \frac{\gamma_{\theta_v} \beta_1 h}{1 + \alpha\beta_1} \tag{5}$$

where $\gamma_{\theta_v}$ is the mean virtual potential temperature lapse rate above the ML and $\alpha$ is a
subsidence-dependent parameter whose likely maximum range is between 1 and 2.
For $w_s = 0$, $\alpha = 2$; and for $dh/dt = 0$ (i.e. $w_e + w_s = 0$), $\alpha = 1$. Substituting eq. (5) into



(4), we get

$$w_e = \frac{(1+\alpha\beta_1)}{\gamma_{\theta_v} h} \overline{(w'\theta_v')}_s$$

(6)

Therefore, $w_e$ is directly proportional to $\overline{(w'\theta_v')}_s$ and inversely proportional to $h$ and
$\gamma_{\theta_v}$.
For the four LES experiments (CON, $1.2\rho_{CON}$, $1.4\rho_{CON}$, $1.7\rho_{CON}$) with varying $\rho$,
we can confirm that $\overline{(w'\theta_v')}_s$ is inversely proportional to $\rho$ with constant sensible heat
flux $H = \rho c_p \overline{(w'\theta_v')}_s$ as shown in Figure 2(c). As shown in Figure 2 (a)-(d), with
increasing air density, the increase in $h$ with time is delayed, and there are also
obvious delays in the time of the first occurrence of cumulus clouds and cloud core.
The increase in $h$ can be divided into 3 stages: 1. $h$ increases slowly with time when $h$
is less than 0.5 km; 2. the growth rate of $h$ obviously increases between about 0.5 km
and 1.5 km; 3. the growth rate of $h$ slows down when $h$ exceeds 1.5 km. In the first
stage the strong inversion layer at the top of the nighttime stable boundary layer (SBL)
gradually erodes due to surface heating. Compared to the high $\rho$ case, the strong
inversion layer vanishes faster for the low $\rho$ case due to larger $\overline{(w'\theta_v')}_s$. During the
second stage, the increase of $\overline{(w'\theta_v')}_s$ and decrease of $\gamma_{\theta_v}$ lead to larger $w_e$ and thus the
growth rate of $h$. This phenomenon is more obvious for low $\rho$ over the TP. In the final
stage, $h$ increases relatively slowly over time, but $h$ is significantly larger for small $\rho$
than for large $\rho$. There are no significant differences in $RH_0$ and relative humidity
above $h$ for the four LES experiments (CON, $1.4\rho_{CON}RH0.05$, $1.4\rho_{CON}RH0.15$,
$1.4\rho_{CON}RH0.3$, Figure omitted). Therefore, we conclude that larger $RH_h$ and more
favorable conditions for saturation occur for small $\rho$ compared to large $\rho$.
4.2 Penetrative convection at the top of a growing mixed layer with varying air
density
Penetrative convection at the top of the ML can result in cumulus formation (e.g.
Stull, 1988). A forced cloud will form when a thermal reaches the lifting condensation
level (LCL), but the top of the forced cloud does not reach its level of free convection
(LFC). Condensation and latent heat release are insufficient to produce positive
buoyancy within the forced clouds, so they remain shallow and undeveloped. Active
clouds have positive buoyancy when the updraft reaches the LFC.
Decreasing $\rho$ leads to an earlier appearance of cloud cores. With increasing $\rho$, $h$
corresponding to the appearance of active cloud (the fraction of cloud core $a_{cc} >$
0.01%) gradually increases as shown in Figure 2(a). With the same $h$, the $RH_h$
corresponding to $h$ is basically the same for the four LES experiments (CON, $1.2\rho_{CON}$,





$1.4\rho_{CON}$ , $1.7\rho_{CON}$). As a result, the differences in the appearance of active cloud
among the four experiments can be considered independent of $RH_h$ in this case. Zhu et
al (2002) defined local CBL height $h_{local}$ as the height where the gradient of any
conserved variable starts to change dramatically. Here the determination of $h_{local}$ is
consistent with $h$, and the penetration depth $d_t$ at any location is defined as the
difference between $h_{local}$ and $h$. Here $a_{cp}$ and $a_{ccp}$ are the projection of the three
dimensional cloud and cloud core fields on the XY plane, respectively. Figures 3(a)
and (b) show that the proportion of the area of deeper $d_t$ ($d_t > 0.3$ km) for small $\rho$ is
significantly larger than for large $\rho$ (25.67% versus 3.05%). There is a good
correspondence between the horizontal distribution of $a_{cp}$ and larger $d_t$, and for small $\rho$
a cloud core forms only at the location of maximum $d_t$ .

When thermals overshoot into the inversion layer, they become negatively

buoyant and decelerate. Compared to the large $\rho$ case, stronger local ascending
motion appears (Figure3 (c) X ≈ 5.2 km) for the small $\rho$ case, corresponding to larger
overshoot, and greater probability of the air parcel reaching LCL and LFC . Both
cloud cover and cloud cores appear in the area of strong ascending motion above the
ML. Thus, the areas of cloud fractions $a_{cp}$ and $a_{ccp}$ for small $\rho$ are larger than for large
$\rho$.
4.3    Cloud fraction, vertical velocity and mass flux for varying air density

Three mass flux schemes were used for LES of the Small Cumulus Microphysics

Study (SCMS) and Atmospheric Radiation Measurement (ARM) cases: 1. moist static
energy convergence closure; 2. Convective available potential energy (CAPE)
adjustment; 3. a subcloud convective velocity scaling scheme. The details of the first
two schemes are described in Gregory et al (2000). The third scheme was first
proposed by Grant (2001) who used turbulent kinetic energy arguments to link the
cloud base mass flux to the convective vertical velocity scale of the ML. The three
schemes results were compared with the LES results of Negger et al (2004). In
general, the third scheme showed a best agreement with LES results in the
reproduction of the diurnal variation of the mass flux at cloud base in shallow
cumulus convection. However, the algorithm proposed by Grant (2001) produces
cloud base mass fluxes too early due to lack of cloud core information. Negger et al
(2004) added cloud core fraction to solve this problem. We discuss the effects of air
density on cloud or cloud core fraction, vertical velocity and mass flux, and the
applicability of the third scheme.

Cuijpers and Bechtold (1995), Neggers et al (2006) and van Stratum (2014)

indicated that the cloud fraction at the top of the ML can be estimated by the average



saturation deficit $\left(q_{t;h} - \overline{q}_{s;h}\right)$ and the spatial moisture distribution that can be
described by specific humidity variance $\sigma_{q;h}$. The $\left(q_{t;h} - \overline{q}_{s;h}\right)$ are the differences
between the specific humidity $q_{t;h}$ and the saturation specific humidity $\overline{q}_{s;h}$ at the
ML top. The parameterization of the maximum cloud fraction at cloud base $a_c$ is
assumed to be:
$$a_c = 0.5 + \alpha \arctan\left(\beta \frac{\left(q_{t;h} - \overline{q}_{s;h}\right)}{\sigma_{q;h}}\right) \tag{7}$$

where the constants $\alpha = 0.36$, $\beta = 1.55$ are used to fit this function to LES results as
proposed by Cuijpers and Bechtold (1995). As shown in Figure 4 (a) and (b), for CON
and $1.4\rho_{CON}$, we see that although the relationship between $a_c$ and $(q_{t;h} - \overline{q}_{s;h})/\sigma_{q;h}$
basically satisfies eq. (7), it also overestimates $a_c$ relative to LES results, especially
for smaller $a_c$ ($a_c < 5\%$). The relatively large $\sigma_{q;h}$ for small $\rho$ is an important reason
for the high frequency of occurrence of larger $a_c$, while large $\sigma_{q;h}$ is associated with
the entrainment of drier air between the moist thermals. Although $a_c$ generally
increases with increasing $RH_h$, relatively large $\sigma_{q;h}$ is an indispensable condition for
the appearance of larger $a_c$. When $\sigma_{q;h} < 0.2 \text{ g kg}^{-1}$ and $a_c < 5\%$, $a_c$ is significantly less
than that calculated from eq. (7). In the cumulus developmental stage, the thermals
with strong ascending motion transport more moisture from the subcloud layer into
the cloud layer, thereby significantly increasing $a_c$. For the single purpose of
introducing the first-order feedbacks between core fraction and mass flux, Negger et
al (2004) temporarily simplified the relationship between $a_c$ and $a_{cc}$ to linear relation,
$$a_{cc} = \kappa a_c, \tag{8}$$

where $\kappa$ is a constant ($\kappa = 0.3$). In fact, Negger et al (2004) considered $\kappa$ should
be a variable rather than a constant. The factors that can affect the variation of $\kappa$
should be analyzed and discussed, and on this basis we can build a more sophisticated
parameterization of the core fraction. As noted above, Figure 4 (c) and (d) also show a
similar trend in that both the areas of $a_c$ and $a_{cc}$ for small $\rho$ are larger than for large $\rho$.
However, with increasing $\rho$, $\kappa$ decreases from 0.27 to 0.03.
On the other hand, LeMone and Pennell (1976) observed that cumulus clouds
often are deeply rooted in the subcloud layer as dry thermals. Based on the above
findings, Neggers et al (2004) proposed a relationship between the convective
velocity scale of the subcloud layer $w_*$, and $w_{cc}$:
$$w_{cc} \approx \lambda w_* = \lambda \left(\frac{gh}{\Theta_v^0} \overline{\left(w'\theta_v'\right)_s}\right)^{1/3}, \tag{9}$$





where $g$ is the gravitational acceleration, $\Theta_v^0$ is the average virtual temperature of the
subcloud layer, and $\lambda$ is a proportionality factor. Neggers et al (2004) and Ouwersloot
et al (2014) proposed that $\lambda \approx 1$, while van Stratum et al (2014) estimated $\lambda \approx 0.84$
based on results from the Dutch Atmospheric LES.

As expected, $w_{cc}$ increases with increasing $w_*$ as shown by the results of three

LES experiments (CON, $1.2\rho_{CON}$ and $1.4\rho_{CON}$) in Figure 4(d). However, with
increasing $\rho$, the rate of reduction of $w_{cc}$ is much faster than $w_*$, and $\lambda$ decreases from
0.7 to 0.46. The deviation between the results of our sensitivity experiments and
previous research increase for increasing $\rho$. The cases studied by Neggers et al (2004)
and van Stratum et al (2013) are at low altitudes, but the results of $\lambda$ and $\kappa$ at high
altitudes in this study are closer to previous research rather than those at low altitudes.
There seems to be a contradiction between the two, and it seems worthwhile to
discuss the reason for the large deviation of $\lambda$ and $\kappa$ for different values of $\rho$. The
results for $1.7\rho_{CON}$ are not given in Figure 4 due to very small $a_{cc}$. We found in our
sensitivity experiments that the values of $\lambda$ and $\kappa$ are determined by the strength of
ascending motion within the thermal characterized by $w_*$ and subcloud layer moisture.
As shown in Figure 4(e) and (f), when the ascending flow within the thermal reaches
the LCL in the drier subcloud layer, there is a relatively small probability of air
parcels reaching the LFC due to small latent heat release. In this case the cloud core
buoyancy at cloud base height,
$$B_{cc} = \frac{g}{\overline{\theta}_v}\left(\theta_{v,cc} - \overline{\theta}_v\right),\qquad(10)$$

$B_{cc}$ is also small (Figure omitted), where $\theta_{v,cc}$ and $\overline{\theta}_v$ are average potential temperature
of the cloud core and all grid points at cloud base height, respectively. This results in a
more rapid decrease in $\lambda$ and $\kappa$ relative to a moister subcloud layer. Larger $a_{cc}$, $w_{cc}$,
and $B_{cc}$ generate stronger updrafts within the thermals for small $\rho$, which favors the
further development of cumulus as shown in Figure 4(c) and (d). Small $\rho$ to some
extent compensates for the drier subcloud layer. In addition, we found that the
deviation from multiple sensitivity tests between the values of $a_{cc}$, $w_{cc}$, $\lambda$ and $\kappa$ for
varying $\rho$ increases with decreasing relative humidity in the subcloud layer. The water
vapor case for SCMS is moister than that of ARM. Therefore, the reason that Neggers
et al (2004) found from LES that $\lambda \approx 1$ for the SCMS case can at least be partly
explained.

5   Discussion

Water vapor is relatively abundant over ECMR in summer. However, observations





indicate that high $LCC$ occurs mainly over the mid-eastern TP rather than ECMR
during summer. Statistical results from ERA-Interim reanalysis data indicate that $LCC$
might still be greater than 35% north of 30 ̊N over the mid-eastern TP for small $RH_{2m}$
($RH_{2m} < 70\%$), and this is not the case at low altitude. The surface buoyancy flux over
the TP is obviously larger than that over lower altitude in eastern China. This density
effect is demonstrated with a simple mixed-layer model in Appendix A and further
confirmed by LES with the same initial profiles of $T$, $RH$ and surface layer turbulent
fluxes but different values of $\rho$. That is, reducing $\rho$ increases thermal turbulence and
overshooting, which increases the probability of air parcels reaching the LCL and
LFC and thus the growth rates of $h$ and $RH_h$, which favor cloud formation. Stronger
ascending motions transport more moisture from the subcloud layer into the cloud
layer, and $w_{cc}$ and $a_{cc}$ also increase. The results also indicate that the values of $\lambda$ and $\kappa$
are determined by the strength of ascending motion within the thermal that can be
characterized by $w_*$ and subcloud layer moisture. Previous research for a drier
subcloud layer has suggested that $\kappa = 0.3$ and $\lambda \approx 0.84$, mainly because the smaller
latent heat release reduces cloud core formation, which causes a significant decrease
in $\lambda$ and $\kappa$. The values of $\lambda$ and $\kappa$ for small $\rho$ are significantly larger than for large $\rho$,
especially with a drier subcloud layer. Based on the above analysis, we find that
smaller $\rho$ over the TP lead to stronger thermal turbulence which favors the formation
and development of convective cloud as demonstrated by the climate statistics of the
$LCC$ in summer over China as shown in Figure 1. Here we analyzed the effect of only
air density on convection and cloud formation. Further studies need to be conducted
on the effects of other factors (e.g. vertical wind shear and complicated heterogeneous
terrain).
6   Conclusions
The cumulus extent and thermal turbulence over the TP are larger than those
over the eastern plain of China. When the relative humidity at 2 m height over the TP
is less than 70%, the coverage of low clouds still exceeds 35%, which is rare over the
east China plain.
For the same surface sensible heat flux over the TP and the elevated plain, the
buoyancy flux over the plateau is larger than over the plain due to the smaller air
density which increases the mixing layer height and the relative humidity at the top of
the mixed layer. This favors the formation of cumulus clouds over the plateau and
increases the probability of the air mass reaching the lifted condensation level and the
level of free convection. More water vapor is transported into the clouds from the





subcloud mixed layer, and the rate of cumulus growth is increased.
The values of $\lambda$ and $\kappa$ in the subcloud convective velocity scaling mass flux
scheme decrease with lower surface relative humidity and weaker thermal turbulence
in the subcloud layer, and thus the values obtained from previous studies may not be
applicable to a drier subcloud layer or weak thermal turbulence cases.

354                                    Appendix A

A SIMPLE MODEL FOR INCORPORATING DENSITY EFFECTS IN THE

356          GROWTH RATE OF THE CONVECTIVE BOUNDARY LAYER

Here we present a simple model to demonstrate that a decrease in surface atmospheric
density in the clear convective boundary layer (CBL) increases the growth rate of the CBL
depth $h$. The model illustrates how the same surface sensible heat flux $\rho c_p (\overline{wT})_0$ results in an
increasing surface buoyancy flux, $(g/T) (\overline{wT})_0$, with decreasing air density $\rho$. The
development utilizes the model of Tennekes (1984) that predicts CBL height $h$ and
magnitude of the temperature jump $\Delta T$ across the CBL top assuming no mean vertical
motion in the CBL and horizontal homogeneity. We assume a dry well-mixed CBL so that
$\gamma_d - \gamma = 0$, where $\gamma = - dT/dz$ and $\gamma_d = 9.8 \mathrm{K\ km^{-1}}$ is the dry adiabatic lapse rate, throughout
the entire CBL and $\Delta T$ is assumed to be discontinuous; that is, we assume the entrainment
layer has zero thickness. Above $h$, the free troposphere is assumed to have a constant potential
temperature lapse rate $\gamma_\theta = \gamma_d - \gamma = d\theta/dz$. This model has been widely used and generally is
successful in predicting reasonable values for $h$ and $\Delta T$ during the rapid growth phase of the
daytime CBL at least up to early afternoon and before clouds form with moderate or less
mean wind speeds and approximately barotropic conditions.
The model equations start with a relation for the temperature flux at $h(t)$, $(\overline{wT})_h$,
which is equal to the rate at which heat is entrained into the CBL. This yields
$$-(\overline{wT})_h = \Delta T \frac{dh}{dt}. \tag{A1}$$

The net rate of change of $\Delta T(t)$ is given by:
$$\frac{d\Delta T}{dt} = \gamma_\theta \frac{dh}{dt} - \frac{\partial \overline{T}}{\partial t}, \tag{A2}$$

where $\overline{T}$ is the mean mixed-layer temperature. The rate of change of $\overline{T}$ is given by
$$\frac{\partial \overline{T}}{\partial t} = -\frac{\partial (\overline{wT})}{\partial z} = \frac{(\overline{wT})_0}{h} - \frac{(\overline{wT})_h}{h}, \tag{A3}$$

since $(\overline{wT})$ is a linear function of height.
Substitution of (A3) into (A2) yields





$$h\frac{d\Delta T}{dt} = \gamma_\theta h\frac{dh}{dt} - (\overline{wT})_0 - \Delta T\frac{dh}{dt}. \tag{A4}$$


This can be rearranged to


$$\frac{d(h\Delta T)}{dt} = \gamma_\theta h\frac{dh}{dt} - (\overline{wT})_0. \tag{A5}$$


We integrate (A5) from $t = 0$, which is the start of solar heating in the morning, to the time $\tau$ at which we obtain a measurement of $h$:


$$h\Delta T - h_0\Delta T_0 = \frac{1}{2}\gamma_\theta(h^2 - h_0^2) - H_\tau/\rho c_p, \tag{A6}$$


where $H_\tau$ is the integrated sensible heat flux,


$$H_\tau = \rho c_p \int_0^\tau (\overline{wT})_0 dt. \tag{A7}$$


Thus, we have a relationship involving two unknowns: $\Delta T$ and $h$. To reduce this to one unknown, we introduce the relation


$$(\overline{wT})_h = -\beta_1(\overline{wT})_0, \tag{A8}$$


where the entrainment coefficient $\beta_1$ is assumed to be an empirical constant that has been estimated by multiple numerical and observational studies (e.g. Sullivan et al., 1998). Next we modify the first term in (A4) and substitute (A1) and (A8) into this expression to obtain


$$h\frac{d\Delta T}{dt} = h\frac{d\Delta T}{dh}\frac{dh}{dt} = \beta_1 h\frac{d\Delta T}{dh}\frac{(\overline{wT})_0}{\Delta T}. \tag{A9}$$


We then substitute (A9) into (A4) which yields


$$h\frac{d\Delta T}{dh} + (1+\frac{1}{\beta_1})\Delta T - \gamma_\theta h = 0. \tag{A10}$$


The solution to this is


$$\Delta T h^{\frac{1+\beta_1}{\beta_1}} = \frac{\gamma_\theta}{2+1/\beta_1}h^{(1+\frac{1+\beta_1}{\beta_1})} + C. \tag{A11}$$


Where $C$ is a constant. In order to give an estimate of the expected magnitude and functional dependencies in (A11), we insert a typical value for $\beta_1$ of 0.2 (e.g. Sullivan et al., 1998). Substituting this into (A11), we obtain


$$\Delta T h^6 = \frac{\gamma_\theta}{7}h^7 + C. \tag{A12}$$


To evaluate $C$, we consider that in the morning at $t = 0$, $h(0)$ is very small compared to later in the day, while $\Delta T$ changes much less, so that $C$ must also be small compared to $h(\tau)$, especially since $h$ is taken to a very large power, and thus can be neglected as soon as $h$ becomes several times $h_0$. Therefore,


$$\Delta T \simeq \frac{\gamma_\theta h}{(2+1/\beta_1)} = \frac{\gamma_\theta h}{7}. \tag{A13}$$


If we again assume $h_0$ and $\Delta T_0$ are small, from (A6) we have


$$h\Delta T \simeq \frac{1}{2}\gamma_\theta h^2 - H_\tau/(\rho c_p). \tag{A14}$$


Substituting (A13) into (A14),






$$h^2 \simeq 2H_\tau \frac{(2\beta_1+1)}{\gamma_\theta \rho c_p}. \tag{A15}$$

This gives us a relation to estimate the CBL height at a specific location given the integrated
temperature flux from the initiation of surface heating in the morning to a time $\tau$ presumed
to be before mid-afternoon when the surface heating has dropped significantly from its
mid-day maximum. Alternatively, it may also be possible to use (A15) to estimate $H_\tau$ if
$h$ and $\gamma_\theta$ are known.
We now apply (A15) to estimate the effect of air density $\rho$ on $h$. Here we assume two CBL
heights: one at sea level $h_0$ and the other at $h$. We further assume that the integrated sensible
heat flux at each location is the same, that is, $H_\tau = H_{\tau 0}$. From the hydrostatic equation, we
have
$$\frac{dp}{p} = -\frac{g}{R_d T}dz = -\frac{g}{R_d(T_0+\gamma z)}dz, \tag{A16}$$

where $z$ is the surface elevation above sea level, $R_d = 287.06$ J kg$^{-1}$ K$^{-1}$ is the dry air gas
constant, and $g = 9.807$ m s$^{-2}$ is the gravitational acceleration. From the ideal gas law,
$$\frac{d\rho}{\rho} = \frac{dp}{p} - \frac{dT}{T}. \tag{A17}$$

Then, substituting (A16) into (A17) we obtain
$$\frac{d\rho}{\rho} = -\frac{g}{R_d}\frac{dz}{(T_0+\gamma z)} - \gamma\frac{dz}{(T_0+\gamma z)} = -\left(\frac{g}{\gamma R_d}+1\right)\frac{\gamma dz}{(T_0+\gamma z)}. \tag{A18}$$

Integrating from $z = 0$ to $z$,
$$\frac{\rho}{\rho_0} = \left(\frac{T_0+\gamma z}{T_0}\right)^m, \tag{A19}$$

where $m = -\left(\frac{g}{\gamma R_d}+1\right)$.
Substituting $h$ at height $z$ and $h_0$ at height $z_0 = 0$ into (A15) and taking the ratio of the
heights, we have
$$h/h_0 = (\rho/\rho_0)^{-1/2} = \left(\frac{T_0+\gamma z}{T_0}\right)^{-m/2}, \tag{A20}$$

As a demonstration of the impact of $\gamma$ on $h/h_0$ we show in Figure A1 how $h/h_0$ changes
with $z$ for different values of $\gamma_\theta$ starting with the lowest level in Table B1 of Appendix B and
decreasing to a value close to that of the second level, i.e.: $\gamma_\theta = \{20, 14, 8, 2\}$K km$^{-1}$ or $\gamma =$
$\{-10.2, -4.2, 1.8, 7.8\}$ K km$^{-1}$. Here we assume that the entire layer through which we
calculate $h$ has the same $\gamma_\theta$, and standard atmosphere values of $T_0 = 288.16$K and $\rho_0 =$
1.225 kg m$^{-3}$. We see a strong dependency of $h/h_0$ on $\gamma_\theta$; for example, $h/h_0$ is almost 20%
larger than its sea level value at 4 km elevation for $\gamma_\theta = 20$ K km$^{-1}$ and more than 30% larger
for $\gamma_\theta = 2$ K km$^{-1}$.



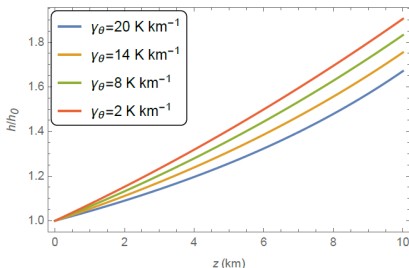


Figure A1. Ratio of the CBL height at an elevation $z$ versus height at sea level for $\gamma = 20$
(blue), $= 14$ (orange), $= 8$ (green), $2$ (red) K km$^{-1}$.

Appendix B
THE LES MODEL DESCRIPTION AND SIMULATION SETUP, AND A
COMPARISON BETWEEN THE OBSERVATIONS AND THE LES
The LES experiments discussed here were performed with the Dutch
Atmospheric LES (DALES) (Heus et al. 2010) using the Deardorff subgrid-scale
closure (Deardorff, 1973) and a second-order advection scheme for scalars,
momentum, and turbulence kinetic energy. We used the radiation scheme proposed by
Fu and Liou (1993) and Pincus and Stevens (2009), and a simple ice microphysics
scheme (Grabowski, 1998) that considers the impact of the relatively low
temperatures over the TP on ice phase microphysical processes. A resolution of 6.4
km x 6.4 km x 6.0 km with 256 x 256 x 150 grid points is used, with a total
integration time of 50400 s. Zhang et al (2017) pointed out that an effective way to
simulate shallow cumulus is by building a composite modeling case (average values
of multiple "golden days"). Using this method, we attempted to construct the initial
profiles, surface turbulent fluxes and large-scale forcing in the control experiment
(CON) by selecting nine shallow cumulus days at Nagqu over TP. In order to reduce
the differences between the LES and the observations, 9-day means were slightly
modified. We adopted the method proposed by van der Dussen et al (2013) to
construct the initial profiles of virtual potential temperature $\theta_v$ and specific humidity
$q_T$ by dividing them into 3 linear segments,
$$\varphi(z) = \begin{cases} \varphi_1 + z\Gamma_{\varphi_1} & 0 \text{ km} < z \le 0.5 \text{ km} \\ \varphi_1 + 0.5\Gamma_{\varphi_1} + (z - 0.5)\Gamma_{\varphi_2} & 0.5 \text{ km} < z \le 4 \text{ km}, \quad \text{(B1)} \\ \varphi_1 + 0.5\Gamma_{\varphi_1} + (4 - 0.5)\Gamma_{\varphi_2} + (z - 4)\Gamma_{\varphi_3} & 4 \text{ km} < z \le 6 \text{ km} \end{cases}$$


with the constants given in Table B1. Where $\varphi \in \{q_T, \theta_v\}$ are the total specific
humidity and the virtual potential temperature, respectively.

Table B1 Values of the constants which are used to describe the initial profiles shown
in Figure S1

| $\Gamma_{q_{T1}}\left(\text{g kg}^{-1}\ \text{km}^{-1}\right)$ | $\Gamma_{q_{T2}}\left(\text{g kg}^{-1}\ \text{km}^{-1}\right)$ | $\Gamma_{q_{T3}}\left(\text{g kg}^{-1}\ \text{km}^{-1}\right)$ | $q_{T1}\left(\text{g kg}^{-1}\right)$ |
|---|---|---|---|
| -2.4 | -0.89 | -0.2 | 4.8 |
| $\Gamma_{\theta_{v1}}\left(^{\circ}\text{K km}^{-1}\right)$ | $\Gamma_{\theta_{v2}}\left(^{\circ}\text{K km}^{-1}\right)$ | $\Gamma_{\theta_{v3}}\left(^{\circ}\text{K km}^{-1}\right)$ | $\theta_{v1}\left(^{\circ}\text{K}\right)$ |
| 20 | 2.29 | 5.5 | 320 |

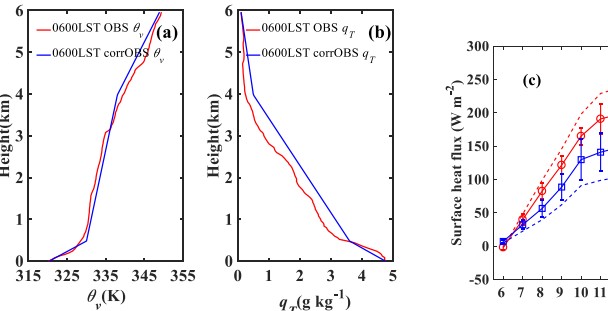

Figure B1 Vertical profiles of (a) $\theta_v$ and (b) $q_T$ at 06:00 LST. The red and blue
lines are the observations and LES initial profiles, respectively. (c) The solid red and
blue lines are the nine-day averaged sensible heat flux $H$ and latent flux $LE$,
respectively, calculated by the eddy covariance method. Error bars represent standard
deviations. The dashed red and blue lines are the "corrected" sensible heat fluxes
$corrH$ and latent fluxes $corrLE$.
As shown in Figure B1, the diurnal maximum of both the sensible heat flux $H$
and the latent flux $LE$ at Nagqu occur at roughly the same time (12:00 LST), and $H$ is
larger than $LE$ during the daytime. However, compared to the radiosonde observations
at 12:00 LST and 18:00 LST, we find poor agreement between the LES results and
the observations when we directly use the $H$ and $LE$ data calculated by the eddy
covariance method without any corrections. The comparison results show that within
the boundary layer $q_T$ is overestimated by 2 g kg$^{-1}$ while $\theta_v$ is underestimated by 4 °K.
To adddress this, we increased $H$ by 20%, and decreased $LE$ by 25%; we call these
corrected values $corrH$ and $corrLE$.
Figures B2(a) and (b) show that the geostrophic wind direction changes
counterclockwise from southeast in the surface layer to northwest in upper levels in
response to the cold advection. As shown in Figures B2(d) and (e), from 06:00 LST to
16:00 LST, the cooling rate caused by weak cold air advection at all levels generally
did not exceed 1 °K day$^{-1}$, and dry advection below 450 hPa was about 0.5 g kg$^{-1}$
day$^{-1}$; thus, temperature and moisture advection were negligible. As shown in Figure




B2(c), the vertical temperature and moisture transport due to large scale subsidence
can result in about 1-2 ˚K warming and 0.5 g kg$^{-1}$ drying after 10 hours.

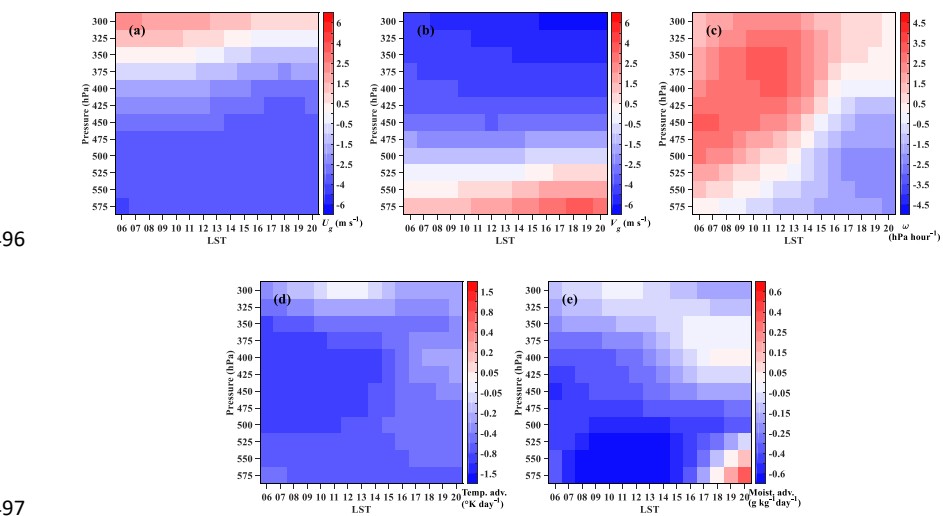



Figure B2 Time–height composite-mean large-scale for geostrophic wind components
(a) $U_g$, (b) $V_g$, (c) subsidence rate $\omega$, (d) temperature advection, and (e) moist
advection for the composite case based on nine days continuous forcing data from
ERA-Interim reanalysis data.
We carried out an LES control experiment (CON) and two sets of sensitivity
experiments: The first set, $1.2\rho_{CON}$, $1.4\rho_{CON}$, and $1.7\rho_{CON}$, have air densities $\rho$ altered
by the factor $r_1$ compared to CON but with the same profiles of $\theta_v$ and relative
humidity. The second set, $1.4\rho_{CON}RH0.05$, $1.4\rho_{CON}RH0.15$, and $1.4\rho_{CON}RH0.3$, have
different relative humidities below 1.5 km, increasing from $RH_{CON}$ to $RH_{CON}$ + (1 -
$RH_{CON}$) x $r_2$ for the $1.4\rho_{CON}$ case. $RH_{CON}$ is the relative humidity for control
experiment (CON), and the values of $r_1$ and $r_2$ are shown in Table B2.
Table B2 Specifications for the LES sensitivity experiments

|  | $r_1$ | $r_2$ |
|---|---|---|
| CON | 1.0 | 0.0 |
| $1.2\rho_{CON}$ | 1.2 | 0.0 |
| $1.4\rho_{CON}$ | 1.4 | 0.0 |
| $1.7\rho_{CON}$ | 1.7 | 0.0 |
| $1.4\rho_{CON}RH0.05$ | 1.4 | 0.05 |
| $1.4\rho_{CON}RH0.15$ | 1.4 | 0.15 |
| $1.4\rho_{CON}RH0.3$ | 1.4 | 0.3 |

Figure B3 shows that the LES model can reproduce the general tendencies and the



diurnal variation of $\theta_v$ and $q_T$ at Nagqu, which indicates that the large-scale forcing
has been correctly specified. There are minor differences between the observations
and the LES; the absolute value of $q_T$ differences generally do not exceed 0.5 g kg$^{-1}$,
and the LES underestimates $\theta_v$ by 1-3 K.

Figure B3 Vertical profiles of (a) $\theta_v$ and (b) $q_T$ at 12:00 LST (solid line) and
18:00 LST (dash line) at the Nagqu site. The red and blue lines represent the observed
profiles from radiosondes and the simulated profiles from CON, respectively.

Data availability. The reanalysis data were from ECMWF (European Centre for
Medium-Range      Weather      Forecasts),      which      is      available      at
https://apps.ecmwf.int/datasets/data/interim-full-mnth/levtype=sfc/                and
https://apps.ecmwf.int/datasets/data/interim-full-daily/levtype=sfc/.      The      original
codes of DALES (Dutch Atmospheric Large Eddy Simulation) were publicly
available at https://github.com/dalesteam/dales. All the original datasets and code
needed to reproduce the simulation results shown in this paper are available upon
request via email: wyj@cma.gov.cn.

Author contributions. YW was responsible for collecting and processing the data, and
manuscript and plot preparation. YW, XX and MZ designed the experiments. YW, XX,
MZ, and DL analyzed the data. YW wrote the paper. DL wrote Appendix A. All
authors contributed to measurements, discussed results, and commented on the paper.

Competing interests. The authors declare that they have no conflict of interest.

Acknowledgements. This study is supported by the Second Comprehensive and
Scientific Investigation and Study of Tibetan Plateau. This study is also supported by
National Natural Science Foundation of China (Grant Nos. 91837310), the National
Natural Science Foundation for Young Scientists of China (Grant Nos. 41805006),
Basic Research Special Project of Chinese Academy of Meteorological Sciences
(2018Y008), and National Natural Science Foundation of China (Grant Nos.
91637102). The National Center for Atmospheric Research is sponsored by the U. S.
National Science Foundation.



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

**Figure**

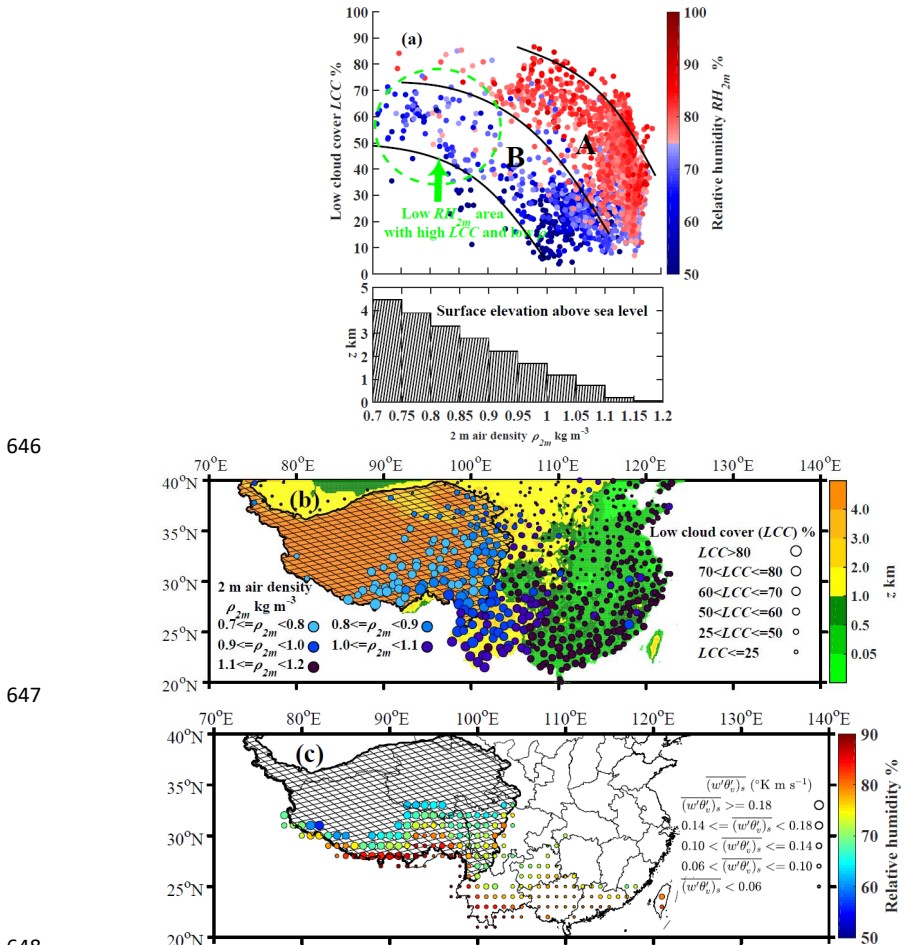




Fig. 1 (a) The relationships among monthly means of $LCC$, $\rho_{2m}$ and $RH_{2m}$ observed by the AWS in summer. The samples are divided into two groups: $RH_{2m} > 75\%$ (red dots) and $RH_{2m} < 75\%$ (blue dots). A region and B region generally correspond to $RH_{2m}$ both greater than and less than 75%, respectively. The histogram shows an approximate relationship between $\rho_{2m}$ and surface elevation above sea level $z$ at the bottom of Figure 1 (a). (b) The spatial distribution of the observed monthly mean $LCC$. (c) The spatial distribution of monthly means of relative humidity and surface virtual potential temperature flux in the surface layer with $LCC$ greater than 35% from ERA-interim data from 9:00 LST to 15:00 LST (3:00 UTC to 9:00 UTC). The TP is the cross-hatched area (altitude > 2500 m) in Figures 1(b) and (c).



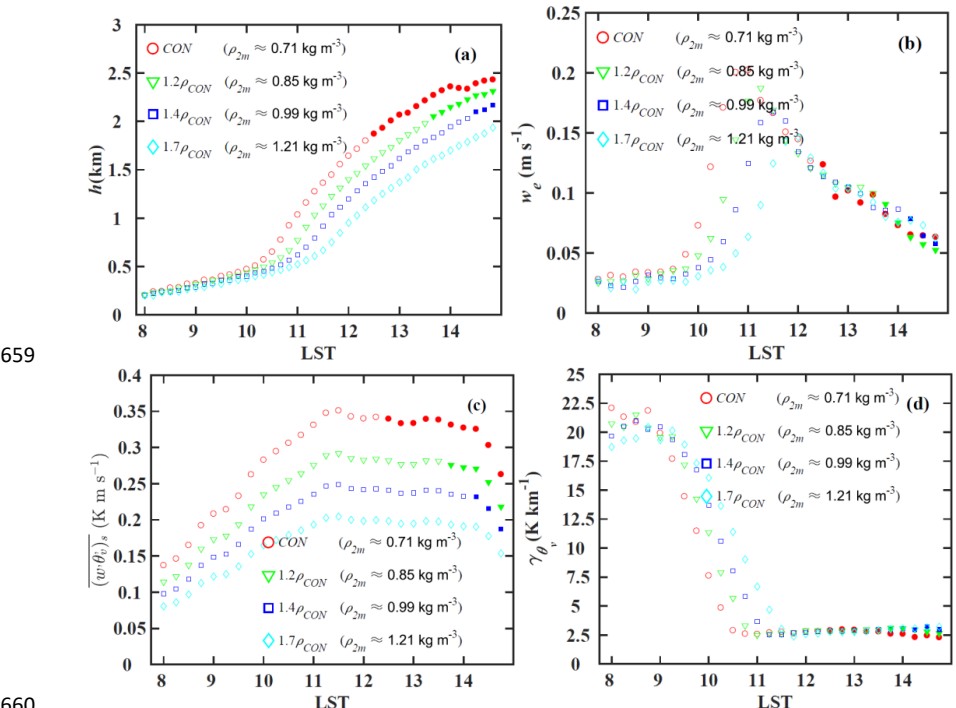

659

660

Fig. 2 The time variations of (a) $h$, (b) $w_e$, (c) surface virtual potential temperature flux, and (d) $\gamma_{\theta_v}$ for four LES experiments (CON, $1.2\rho_{CON}$, $1.4\rho_{CON}$, $1.7\rho_{CON}$) before the early stage of cloud core formation. Solid and hollow symbols represent the presence or absence of cloud core, respectively.

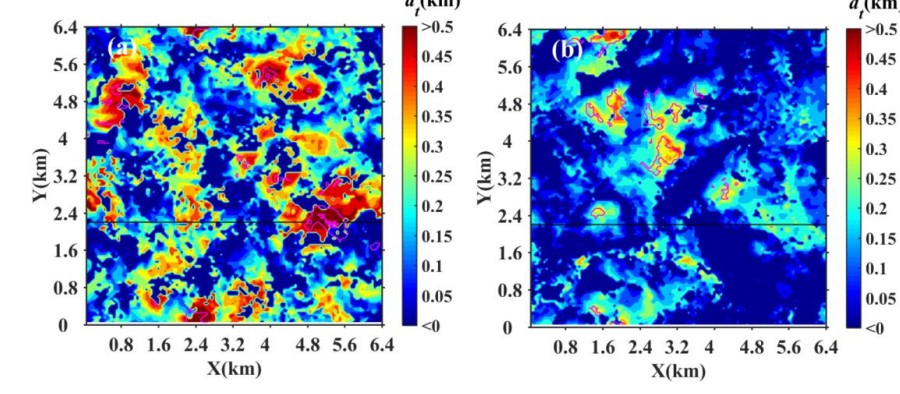


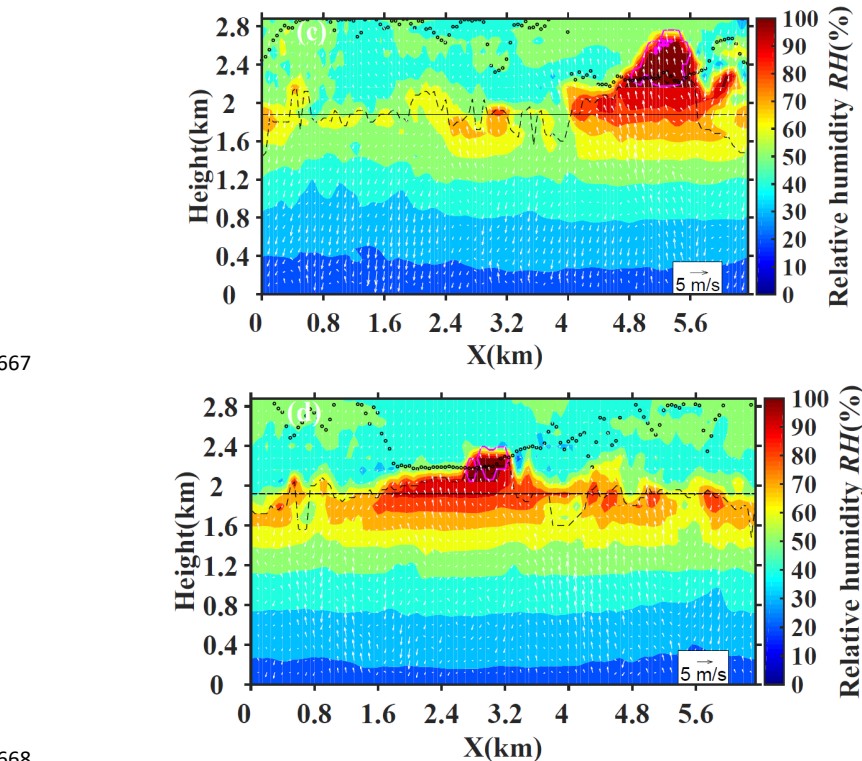



Fig. 3 The horizontal distribution of $d_t$ (color shaded) for $h \approx 1.85$ km for two LES
experiments (a) CON at 12:52 LST (b) $1.4\rho_{CON}$ at 14:10 LST. The area enclosed by
the pink line and solid circles delineate $a_{cp}$ and $a_{ccp}$. The solid straight lines in Figures
(a) and (b) represent the projection of the XZ plane in Figures (c) and (d) on the XY
plane, respectively. The vertical cross-section (XZ-plane) of relative humidity (color
shaded) and wind vectors (X-axis wind speeds are ten times smaller than true values)
for two LES experiments: (c) CON (d) $1.4\rho_{CON}$ obtained along the black solid lines in
Figure 3(a) and (b), respectively. Hollow circles represent the lifting condensation
level of the grids in the X-direction at the height of the mixed layer $z_{lcl}(h)$, and the
pink line and solid circles have the same meaning as in Figure 3 (a) and (b).

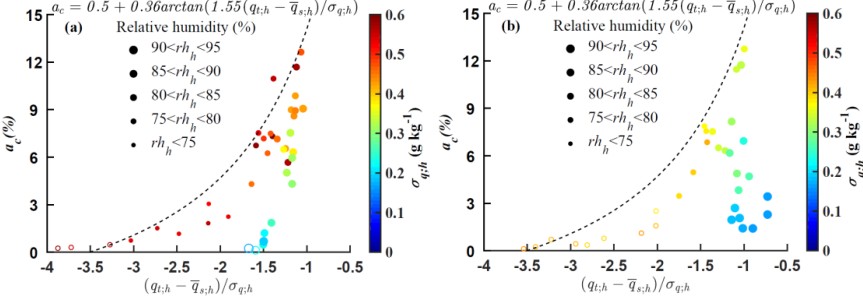




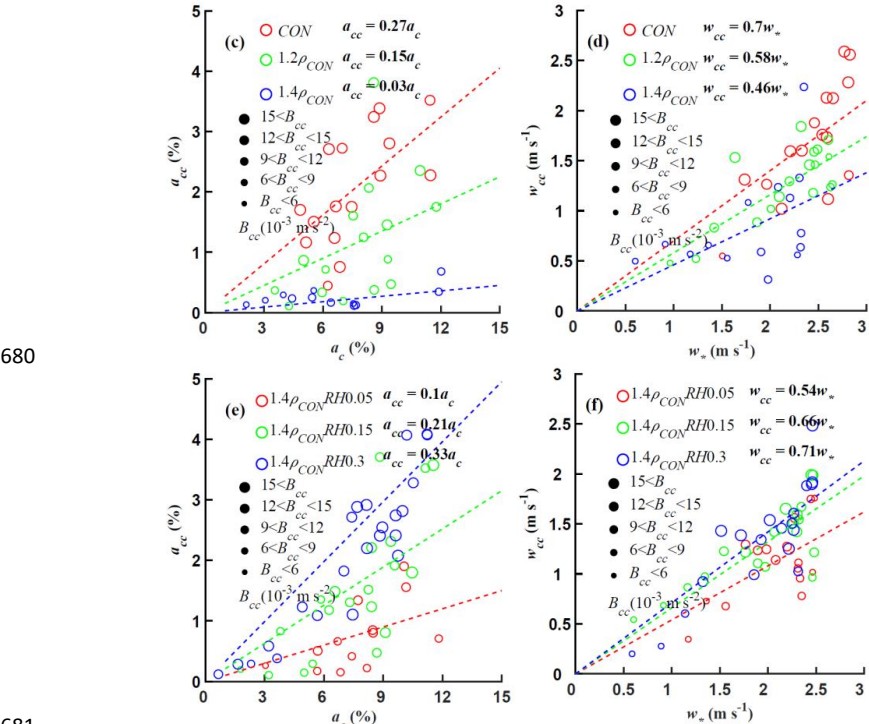

Fig. 4 Scatter diagrams of $a_c$ versus $(q_{t;h} - \bar{q}_{s;h})/\sigma_{q;h}$ for two LES experiments (a) CON (b) $1.4\rho_{CON}$. The black dashed lines are from eq. (7), and the color and the size of the points represent the values of $\sigma_{q;h}$ and $RH_h$, respectively. Scatter diagrams of (c) $a_c$ versus $a_{cc}$ and (d) $w_*$ versus $w_{cc}$ are shown for three LES experiments (CON, $1.2\rho_{CON}$, $1.4\rho_{CON}$). The color identifies the experiments and the size of the points represents the cloud core buoyancy at cloud base $B_{cc}$. Figure 4 (e) and (f) are the same as Figure 4 (c) and (d), respectively, but for the three LES experiments ($1.4\rho_{CON}RH0.05$, $1.4\rho_{CON}RH0.15$, $1.4\rho_{CON}RH0.3$)