# Peer review of "The effect of low density over the "roof of the world" Tibetan Plateau on the triggering of convection"

_Atmospheric Chemistry and Physics, 2019_

## Referee Comment (RC1) · Jun-Ichi Yano (Referee) · 10 Jul 2019

**General Remarks**

First of all, I have to make it clear that I *did not* recommend to post the present manuscript to ACPD. The original submitted main file did not include the appendices referred in the main text, thus I simply suggested that this manuscript should not be considered for a publication until the appendices are included in the main file. Unfortunately, the Editor never returned to me, but simply posted the manuscript without consulting with me.

For the reason just stated above, I did not read through the original incomplete manuscript. In other words, this is the first time that I went through this manuscript.

[Figure]

To be honest, I am not sure whether I would even have agreed to post the present manuscript to ACPD, not to mention a possibility of publishing it in ACP.

In my own reading, the manuscript is badly written: it essentially consists of a random collection of segments of results, which do not present any coherent story. For example, the Appendix A is simply wrong (see below). The paper also quotes various earlier results rather in random manner, by citing randomly chosen references, rather than tracing back original papers, that we are usually expected to do in our citation system.

The title suggests that the paper is considering the effects of low density at high altitudes in boundary-layer convection. An only result I found along this line is (as also explicitly stated in abstract) a rather trivial point that the vertical eddy transport of entropy increases with decreasing density when the surface heat flux (in unit of W/m$^2$) is fixed. However, this statement is meaningful only if the surface heat flux could be considered approximately constant under a change of the density. In fact, their LES simulations are performed by simply assuming this. On the other hand, a simple consideration based on a standard bulk formulation (see below) suggests that it is rather a vertical eddy flux which is invariant with the density. Thus, the whole premise of the work is simply invalidated.

Based on these considerations, I strongly suggest that the present manuscript should not be considered for a publication in ACP. I rather regret that this rather confusing manuscript has even appeared in ACPD.

**General Evaluations of the Work**

The weather of high altitudes is qualitatively different from that of a surface level at which majority of us live. This is a common knowledge for mountaineers, for example. Unfortunately, research of high–altitude weather is relatively scarce. Thus, by addressing this question, the present manuscript could be potentially a very welcome contribution.

It is interesting to note that the present authors try to characterize high altitudes by low densities. Our standard approach takes either a geometrical height or a pressure level as a measure of the altitude. Alternative can be to take the potential temperature, or even the potential vorticity. However, the manuscript never explains well why the authors focus on the density rather than any other physical quantities.

Even worse, reading the abstract, the only part related to the density in their work is that "a decrease in air density enhances the buoyancy flux". As I already stated, this statement is simply trivially true, but only if the surface flux itself is invariant with a change of density. The latter assumption likely fails. Thus, the present work simply loses any relevance.

**Major Issues**

In this section, I am going to discuss major issues with the present manuscript one by one.

*Lack of Logical Coherence:*

The manuscript lacks a logical coherence. The point is already clearly seen in the lead sentence of the abstract, which states that "We study the relationship between convective characteristics and air density over the Tibetan Plateau from the perspective of both climate statistics and large eddy simulations".

Fine, probably, a clever design of LES will give an insight on this problem. Unfortunately, the authors lack with such an ingenuity (cf., Section *LES Experiment Design*). On the other hand, it is totally unclear how this question could even be addressed by any statistical analyses of climate.

The result of the climate analysis is summarized by Fig. 1(a). Here, the horizontal axis is the air density. However, it is clear that the air density can be replaced by any other quantities that characterize an altitude of the ground. Thus, Fig. 1(a) does not demonstrate an important role of surface air density to the problem in any manner.

[Figure]

Interactive
comment

Worse than that, the discussion of the figure does not lead to any questions to be resolved by further analyses. At least, I missed to read such a clear statement in the text.

It seems to me that the figures of the present manuscript are presented in random order, and text could be read equally well even by re–shuffling the order of figures in any different manner. In this manner, the manuscript totally lacks with a clear logical order of a presentation. At least, I fail to follow such a logical sequence.

Most seriously, the manuscript totally fails to explain why the authors focus on the effect of low density with high altitudes, rather than any other attributes (e.g., pressure, potential temperature, etc) of a high altitude. In other words, even a choice of a subject of the study is just a random choice without any justification.

*Geographical Focuses:*

The effect of low density in ABL, if it ever exists in any sensible manner, is clearly a universal question that applies to any ABL regardless of any geographical locations. However, the manuscript strangely focuses exclusively on the Tibetan Plateau without clearly stating a reason for this focus for addressing this very question.

Indeed, the first paragraph of the introduction addresses a uniqueness of the Tibetan Plateau in a very general context of the climate dynamics. However, as far as I can read, the introduction totally fails to explain why we need this particular geographical focus, more specifically, for studying the effect of low density in ABL.

Reading through the manuscript, I even begin to see a nationalistic motivation to study the Tibetan Plateau rather than based on any real scientific motivations. An expression "over the rest of China" at Line 39 suggests that the authors are only interested with Chinese geography. In support of this speculation, Fig. 1 only maps the values over China. This is in spite of the fact that ERA reanalysis that the authors use cover the whole globe. The mnauscript lacks in scientific objectivity, even when it is possible to

maintain it.

**ACPD**

*LES Experiment Design:*

Arguably, a main result of the present study is a set of LES experiments performed for investigating the so–called effect of low density in ABL. However, the actual design is rather naitve, at the best, and most likely very misleading: the surface heat flux, $H$, (given in unit of W/m$^2$) is simplify fixed, and the surface–level density is varied. As a result, the vertical eddy potential–temperature flux (given in unit of Km/s), $\overline{w'\theta'}$, simply increases with a decreasing density. In other words, the authors are *not* examining any genuine effect of density on ABL, but simply changing $\overline{w'\theta'}$, and examining resulting change of the ABL behaviors. The main results presented by Fig. 2 are rather trivial, and I do not see anything particular to report.

*Change of Surface Flux with a Change of Air Density:*

The authors do not provide any sensible argument to justify this experimental design. However, a following simple simply consideration on a principle of the bulk surface–flux formulation suggests that it is rather the vertical eddy flux, $\overline{w'\theta'}$, rather than the heat flux, $H$, itself that remains overall invariant with a change of air density.

A standard bulk formulation writes the surface vertical eddy flux, $\overline{w'\theta'}$, as

$$\overline{w'\theta'} = C u_H(\bar{\theta}^* - \bar{\theta})$$

where $C$ is a bulk coefficient, which generally has only a weak dependence on the environmental state; $u_H$ is a horizontal–wind scale. Thus, if all the other factors are identical, $\overline{w'\theta'}$ remains unchanged with a change of air density, but $H$ decreases with a decreasing air density.

It follows that the present LES design does not reveal anything about the density effect on ABL.

*Appendix A:*
- In the main text, the role of environmental descent, $w_e \neq 0$, is explicitly taken into account (cf., Eq. 2). However, this effect is simply neglected (cf., Eq. A1) without any justifications.

- Throughout the Appendix, the prime sign is missing everywhere for indicating the eddy values. The temperature, $T$, must be replaced by the potential temperature, $\theta$, throughout.

- Eq. (A9) does not make any sense.

- Eq. (A11) is clearly not a solution of Eq. (A10), thus all the subsequent discussions are irrelevant. In fact, a solution for Eq. (A10) is given by Eq. (5) with $\alpha = 2$, as explicitly stated in Zhu et al (2005). [It is obvious that the authors are citing this paper without reading it. Also note that this solution (5) is earlier derived by Betts (1973, see his Eq. 42). However, the authors simply fail to give such a simple credit.]

*Presentation Style:*

See the next section for more specific problems. However, the manuscript generally suffers from the following problems in the presentation style:

- The references are more than often chosen in totally arbitrary manner. It is always important to cite an original source of the idea in citation references. The authors totally dismiss this very basic rule.

- Various equations are quoted rather in an arbitrary manner without any conscious considerations of a relevance.

- Substantial re–writing of the text will be required. I will point selective examples in the next section.

**Specific Comments**

Line 88, EDDYPRO: It is a totally non–essential matter for readers what software the authors have used. However, it is crucial to present how turbulent fluxes are actually computed based on which theory, formula, etc.

Line 107, LCC: Please spell this out

Sec. 3 (Lines 106–120): Please re–write the text to the point. One has to read it dozen times to understand this rather conjugated text to understand what the authors really want to say. As it stands for now, the remaining part of the main text could be read equally well even by totally removing Sec. 3.

Lines 137–146: This paragraph is not linked to any other part of the text. One can simply remove it.

Line 169: Sullivan et al (1998) is just one of many papers quoting an earlier claim with a constant 0.2. This is just an example of arbitrary quotation system that the authors use. The earliest claim of this value that I could dig is Ball (1960). However, the authors must search carefully.

Eq. (5): Indeed this is a valid solution for Eq. (A10) with $\alpha = 2$ when an environmental descent is absent, i.e., $w_e = 0$. However, Zhu et al. (2002) *did not* derive (cf., Line 169) this expression as a general solution for, say, the system with Eq. (2). They merely *suggest* it as a phenomenological generalization of the solution with $\alpha = 2$.

This is an example of the present authors' system of arbitrary quotations of earlier results out of context.

Lines 179–181: This premise is hardly justified, which also makes the present manuscript invalid. A work based on such an ill–posed premise should not be published.

Eq. (7): This is just another example of the present authors' system of arbitrary quotations. This formula is just a curve fit out of an LES, and there is no reason to believe its universality.

Line 311, "Water vapor is relatively abundant over ECMR in summer": This is a very odd manner to start a final section of a paper considering the role of the low density with a high altitude.

Lines 316–317, "This density effect is demonstrated with a simple mixed–layer model in Appendix A": NO

Lines 321–322, "Stronger ascending motions": This is nothing to do with a low density

Lines 452–454, "a simple microphysics scheme (Grabowski et al 1998) that considers the impact of the relatively low temperature over the TP": NO

Line 477, "corrected": Please describe what kind of corrections are made

Fig. 4: plots are so scattered that I do not think that we can draw any sensible conclusions out of them.

---

## Referee Comment (RC2) · Anonymous Referee #3 · 25 Jul 2019

I do not recommend this paper for publication in ACP as it does not meet the essential scientific and presentation requirements. The paper is not well organised and overall difficult to read, so that I was tempted to simply reject it on this basis. The reviewer's work is made harder by the paper structure and lack of clear scientific question or goal. I provide hereafter some comments to justify my decision and possibly help the authors in improving their study, although it is not clear to me if it can be made publishable.

My general advice is to clarify the goals, focus on them, on the most essential points and provide the relevant literature and context, but avoiding constantly going back to lots of literature-related discussions. The paper presentation and the way paragraphs are organised should highlight more the goals and the results rather than have them lost in various elements of literature review distilled in every section of the paper.

[Figure]

General comments:

It is difficult for the reader to understand what are the goals of this study and why it is new and interesting. Most of the paper resembles a catalogue of various informations with little context or perspectives or a literature review but with an unclear positioning of the present study, loosing the reader in lots of information without showing their relevance. There is a lack of hindsight and synthesis work. The description of the figures is usually mixed with too numerous elements from the literature, without much hierarchy in the informations provided.

The authors seem to drag equations from different studies and combine them together without providing any discussion about the underlying assumptions, or even without mentioning the context. Parameters and factors are used out of their context and without proper explanation.

When assumptions are mentioned, they must be justified and it must be clear when they apply and why. The authors must also explain their physical meaning (and their possible limitations), and not simply state with little or no justification "x is negligible" or "we ignore differences between a and b".

The introduction fails in putting the paper in the broader context and does not highlight well enough the state of the art, the reasons for the present study and its precise goals (e.g. question to answer, parametrisation to improve, etc. and why it's needed).

The importance of deep convection is mentioned in the introduction, but then it is unclear how much of the "Low cloud cover" actually comes from deep convective clouds (as the definition used here counts any cloud with "cloud base less than 2.5 km AGL" with no restriction on cloud top height). This must be clarified as deep convection is likely to be part of the observed LCC, but cannot be represented by the LES simulations (due to the domain size).

In several occurrences, sentences are hard to understand as they completely lack of

context so that the reader cannot understand what the authors are referring too. Some typos and English language mistakes / lack of clarity can add to the reader's confusion (for instance LES simulations at "a resolution of 6.4 km x 6.4 km x 6.0 km" or "9-day means" while the total integration time is of 14 hours are both confusing phrases). One of the co-authors is a native English speaker and should then be able to thoroully proof-read the manuscript and help making it clearer and more readable.

Specific comments:

I believe it is not relevant at this stage to give fully detailed specific comments as the manuscript will need profound refactoring. I give hereafter a few specific comments but the list is far from beeing exhaustive, a full review will be needed if a deeply revised, restructured and enhanced manuscript was to be provided by the authors.

l 79 : "interrelation between turbulence and convective motion" : what do you mean here ? This is too vague, and should be better introduced. What has been done before and what is different here.

l70-71 : very vague. Delete or rephrase. It is not very clear from the introduction whar are the "scientific issues"

Fig1a : Are there blue dots under the red ones? It looks like it. Consider using transparency to make the figure more readable and to fully demonstrate your point.

L114-116 : Why is that? Any idea?

L 154: define the cloud cores here. How do you get them from the model?

L157-158 : justify and explain physical meaning of this assumption.

l158-159 : confusing. . . .. Is your model general or only applying to your specific study. Clarification needed. And the sentence must be rephrased.

L 161: how do you know? How where they calculated?

[Figure]

Subsection 4.1: Equations are used out of context without discussing the underlying assumptions and the situation to which they apply. This should be done everytime it is releveant. It is also unclear what is the goal here. Is it to build equation (6)? Under what assumptions and for what purpose?

L199: There is a lack of discussions of the BL structure in the observations and the model, and a figure like Fig B2 should be in the main part of the paper, and compared with the model results. Make it clear that penetrative convection means here dry convection making it through the inversion and then possibly forming a cloud. Or remove this sentence as more explanations follow it anyway

L213-214: unclear, be more explicit.

Section 4.3: this section must be better organised and clarified. More hindsight is needed to avoid only going back and forward from a specific sub-result to one of several cherry-picked models from the literature. What is the reference you are comparing to? Where are the results from the different schemes? If you decide to not show them this must be made very clear.

L 454 − 455 : This surely cannot be the resolution, but must be the domain size. Specify the corresponding resolution and justify the choice of the model top.

Technical :

Only a few comments here as it is not relevant at this state, again not exhaustive at all (cf. My general comments) l133 : "We analyse in detail the results of control experiment ..." : be more precise by adding "In this section ..." l137: it's "water vapor mixing ratio" Eq. (2) : partial derivatives should be used here (no Large scale BL advection is assumed as far as I understood) L 456 : replace by 14 hours, this is more readable.
* * *

---

## Author Comment (AC1) · 30 Jul 2019

**Authors**

The reviewer made the following comment regarding the change of surface flux with a change of air density:

"The authors do not provide any sensible argument to justify this experimental design. However, a following simple simply consideration on a principle of the bulk surface-flux formulation suggests that it is rather the vertical eddy flux rather than the heat flux, $H$, itself that remains overall invariant with a change of air density."

Our response is the following:

A more physical approach to evaluate how the sensible heat flux $H = \rho C_p \overline{w'\theta'}$ at the Earth's land surface varies with air density $\rho$ is to consider the surface energy budget,

$$R_{net} = H + LE + G, \tag{1}$$

where $R_{net}$ is the surface net radiation, $LE$ is the latent heat flux, and $G$ is conduction into the ground. We assume that each of these components is independent of the altitude of the surface, which is approximately the case for identical land surfaces with the same incoming solar radiation and mean atmospheric temperature structure. This is the basis for assuming that $H$, which is proportional to the air density, $\rho$, is independent of altitude. Buoyancy flux, $(g/T)\overline{w'\theta'}$, does not depend on $\rho$. Hence it follows that an increase in altitude means a decrease in $\rho$, and thus an increase in $\overline{w'\theta'}$ to compensate, which then increases the buoyancy flux.

The increase of buoyancy flux with altitude has been documented observationally by Wu et al., 2017) who found that $H$ increases as the

elevation increases from 2000 m to 4000 m over the TP and by Wang et al., 2016 who showed that for unstable stratification in daytime, $(g/T)\overline{w'\theta'}$ over the TP is significantly larger than at lower elevation.

Finally, the derivation of (A10) follows closely the derivation of Tennekes (1973) whom we reference (although with an erroneous date. The date given in the manuscript (1984) is incorrect; it is actually 1973, the same year that Betts independently published a similar relation.). The solution (A11) is a slightly generalized form of his solution that allows for arbitrary values of the entrainment coefficient $\beta$.

References

Tennekes, H.: A Model for the Dynamics of the Inversion Above a Convective Boundary Layer. J. Atmos. Sci., 30, 558-567, 607 https://doi.org/10.1175/1520-0469(1973)030<0558:AMFTDO>2.0.CO;2, 1973.

Wu, G. X., He, B., Duan, A. M., Liu, Y, M., and Yu, W.: Formation and Variation of the Atmospheric Heat Source over the Tibetan Plateau and Its Climate Effects. Adv. Atmos. Sci., 34, 1169–1184, doi: 10.1007/s00376-017-7014-5, 2017.

Wang, Y. J., Xu, X. D., Liu, H. Z., Li, Y. Q., Li, Y. H., Ze, Z. Y., Gao, X. Q., Ma, Y. M., Sun, J. H., Lenschow, D. H., Zhong, S. Y., Zhou, M. Y., Bian, X. D., and Zhao. P.: Analysis of land surface parameters and turbulence characteristics over the Tibetan Plateau and surrounding region, J. Geophys. Res. Atmos., 121, 9540-9560, doi: 10.1002/2016JD025401, 2016.

---

## Referee Comment (RC3) · Jun-Ichi Yano (Referee) · 3 Aug 2019

The lead author posted his response on 30 July to my review comments on 10 July. This is, unfortunately, hardly satisfactory by responding only to two issues, neglecting all the others that I raised. The actual responses on these issues are hardly satisfactory, either.

• Concerning their assumption of a constancy of the sensible heat flux with altitude, the lead author somehow decides to invoke the surface heat budget for justifying it. However, this argument is hardly satisfactory: it essentially claims that since the net radiative heating, $R_{net}$, approximately remains constant with altitude, all the three terms in the right hand side of the budget given by Eq. (1) must also individually remain the

same order. Of course, in reality, these three terms would respond to the net radiative heating differently in different situations, thus such a conclusion does not follow. It is also important to realize that though the solar radiative flux reaching the surface may approximately be constant with altitude, the longwave radiation would change with altitude. Thus the net radiative heating would not be close to constant with altitude in any obvious manner. Clearly, an observational data analysis would be required to make any of such claims. That is totally missing in the present manuscript.

I do not understand why the lead author refers *again* to an observational study by Wang *et al* in the response: such a study does not explain why the vertical eddy heat flux increases with altitude, and more specifically, whether an increase is due to the density effect or not, as the present authors claim.

Strangely, the lead author does not comment on my original argument based on the bulk formulation of the heat fluxes: if their claim is correct, my argument must be disputed.

• Please show an explicit demonstration/derivation that (A11) is actually a solution of (A10), because I strongly doubt it. Recall that, as I stated in the original review comments, Eq. (5) with $\alpha = 2$ is indeed a solution of (A11). It looks like to me, the authors are simply dismissing this simple fact, and trying to "invent" a new solution.

---

## Referee Comment (RC4) · Anonymous Referee #2 · 5 Aug 2019

The manuscript presents an interesting approach to looking at variations in convective activity over large orography, based on the corresponding variations in air density at the surface. The analysis is based on a combination of observational data, global re-analysis, and large-eddy simulation (LES). The results are informative and of sufficient relevance to merit publication in ACP, although I think there are a number of issues which should first be addressed:

[Figure]

**General comments**

1. Although the observation and reanalysis data used are briefly described in Section 2, there appears to be no section describing the LES model in the main paper. Without knowing which model is used, its domain size and resolution and major paramterisations etc., it is impossible to understand the likely impact of model errors in the results presented here. While it's fine to move further detail to an appendix, the basic information should be presented in the main text prior to the presentation of any results.

2. In various places, the variation of behaviour with air density is considered, but it is often not made clear to what extent this means the fixed variation of density due to the orography, or the synoptic variations which may occur at any given location.

3. There is considerable discussion of relationships between density $\rho$ and CBL height $h$. However, it is not obvious that comparing a geometric height/thickness measure across large variations in density is appropriate. Consideration should be given to how the relationships would look with a mass-based measure of thickness (corresponding to a pressure-based rather than height-based vertical coordinate). The same applies to the question of variations in vertical velocity with density – these relationships may look very different between geometric vertical velocity and pressure velocity).

**Specific comments**

**p.3, lines 75–76** Are "very small" horizontal scales of "tens of metres" adequately resolved by the LES configuration used in the study?

**p.4, line 119** Kelvin are not degrees, i.e. the unit is K, not °K.

**p.454, lines 454–455** A resolution on the order of 6km is not an LES model, but in the realm of the highest-resolution global NWP models, or "cloud-system resolving models". Or should this read "A *domain size* of 6.4km × 6.4km × 6.0km. . . "? This would make the horizontal and vertical resolution 250m and 40m respectively, which seems more reasonable, but still unable to resolve well the "tens of metres" scale referred to on lines 75–76.

**Figures 2, 4** The panels should be formatted so that legends do not obscure the actual data points.

**Figure 3** The (a), (b), (c), (d) labels in white can barely be read against the patterns in the actual plots. These labels should probably be moved outside the plot area.

**p.17, lines 520–527** The original datasets used are stated to be available "upon request", rather than being deposited in a readily-accessible archive. I would draw attention to this, but leave it to the editor's discretion whether this is sufficient to meet the journal's data policy without further justification.

---

## Author Comment (AC2) · 22 Aug 2019

We would like to thank the referee for his comments on the manuscript. The following is a point-by-point reply to the comments.

**1. Surface heat flux**

We assumed that the sensible heat flux $H$ is invariant with the density (or elevation) in LES simulations and appendix A. Based on this premise, we discussed the effect of low air density over the Tibetan Plateau (TP) on the formation and development of cumuli. This hypothesis was based on our observational data analysis of the third Tibetan Plateau Experiment (TIPEX III). The analysis showed that for unstable stratification the buoyancy term increases with increasing elevation (see Fig.13 c, .Wang Y. J. et. al., *J. Geophys. Res. Atmos.*, 121, 9540-9560, doi: 10.1002/2016JD025401, 2016.). This implies that the surface heat flux basically does not vary with elevation in unstable stratification.

The bulk transfer formula mentioned by the referee is

$$H = \rho C_p C_h U \left( T_g - T_a \right) \tag{1}$$

Where $\rho$ is the air density, $C_p$ is the specific heat of air at constant pressure, $C_h$ is the bulk transfer coefficient for heat, $U$ is the mean wind speed at 10 m, and $T_g$ and $T_a$ are the surface skin temperature and air temperature at 2 m, respectively. Equation (1) is a semi–empirical formula for calculating $H$. In our case, $U$ increases with increasing elevation over the TP. $C_h$ depends on many factors (e.g. underlying surface conditions, wind speed, stratification, etc); thus $C_h$ is a parameterization for use in numerical models rather than a constant, and $H$ data can be used to estimate $C_h$ from equation (1). We do not discuss

variations of $C_h$ and $U$ for varying elevation.

The referee asserts that $\rho$ (or elevation) has nothing to do with $C_h$, $U$ and $T_g$-$T_a$. If all these variables ($C_h$, $U$ and $T_g$-$T_a$) are fixed, $\overline{w'\theta'}$ would not change with air density and $H$ would then decrease with decreasing $\rho$. However, observations show that $T_g$-$T_a$ significantly increases with increasing elevation over Asia, Australia and South America (Pepin et al., 2005). Xu et al., 2010 observed that if elevation increases from 1000 m to 4000 m over China, $T_g$-$T_a$ increases by an average of about 50% in spring and summer. Wu et al., 2017 came to a similar conclusion by using 73 meteorological stations data over the TP. Although both $T_a$ and $T_g$ decrease with increasing elevation over the TP, $T_a$ decreases faster than $T_g$, so that $T_g$-$T_a$ increases with elevation. It should be pointed out that all the above results derived from daily mean values, the phenomena that $T_g$-$T_a$ increases with an increasing elevation will be more obviously in daytime in summer for the case of our study. It's the surface energy budget that determines the surface heat flux. The temperature difference $T_g$-$T_a$ is a response to the forcing expressed by the surface energy budget.

2. Logical coherence and geographical focuses

Thanks for the referee's suggestion. In order to make the Figure 1 (a) more readable and to fully demonstrate our point, we changed the transparency of the dots (as suggested by referee 3). As we described in our manuscript, Figure 1 (a) shows that more low cloud exists over the high altitude area of the TP with low $\rho_{2m}$ and low $RH_{2m}$ ($50\% < RH_{2m} < 70\%$), and figure 1 (b) indicates that high $LCC$ (low cloud cover) occurs mainly over the mid-eastern TP rather than eastern China monsoon region

(ECMR) during summer. From ERA-Interim reanalysis data, Figure 1 (c) indicates that $LCC$ might be greater than 35% north of 30 °N over the mid-eastern TP for $RH_{2m} < 70\%$, and this is not the case at low altitude. These are only statistical results, so we use large eddy simulation (LES) to partly explain the reason for this phenomenon. In order to clearly state the questions to be resolved by further analyses, and improve the coherence of our manuscript, the corresponding parts of the sentences will be revised. Here we explain the reason why we only map the values over China rather than global. The title of the manuscript is "The effect of low density over the "roof of the world" Tibetan Plateau on the triggering of convection"; thus we mainly focus on the TP and surrounding region. TP has a uniquely large area with an average altitude above 4000 m, a land surface that absorbs a large fraction of incoming solar radiation, and a relative humidity over the mid-eastern TP that is typically larger than 50% in summer. Furthermore, we have extensive data over the TP. However, the results should have general application to regions with similar conditions. Therefore, we plot Figure. S1 and S2 from ERA-interim reanalysis data in East Asia in summer, here the samples include all the grids on land within the latitude and longitude range (70 °E-140 °E, 0 °N-60 °N). Figure S1 (a) shows that $H$ has no obvious correlation with elevation. Both $T_g$-$T_a$ and $\overline{w'\theta'}$ significantly increase with increasing elevation as shown in Figure S1 (b) and (c). The results from ERA-interim data are consistent with our previous data analysis.

The referee asked why we choose density rather than other physical quantities (.e.g. pressure, potential temperature, etc). In fact, all the LES sensitivity tests in our manuscript have taken into account the variations

of pressure and potential temperature for varying air density (or elevation). The LES sensitivity tests ($1.2\rho_{CON}$, $1.4\rho_{CON}$, and $1.7\rho_{CON}$) have the same initial profiles of temperature $T$ and relative humidity $RH$ from the surface to 6 km above the ground level (Sorry, there is a mistake in line 504, it should be temperature $T$ rather than virtual potential temperature $\theta_v$). This manuscript mainly discusses the growth rate of the convective boundary layer and the effect of thermal turbulence on the formation and evolution of cumulus in daytime, so we mainly focus on the effect of surface sensible heat flux especially for varying air density. This is also the reason why we set the same initial profiles of temperature $T$, relative humidity $RH$ and large scale forcing for LES sensitivity tests ($1.2\rho_{CON}$, $1.4\rho_{CON}$, and $1.7\rho_{CON}$). In order to illustrate our point further, the relationships among monthly means of $LCC$, $\overline{w'\theta'}$ and $RH_{2m}$ ($RH_{2m} > 50\%$) from ERA-interim data are also plotted as shown in Figure S2. With the same $\overline{w'\theta'}$, $LCC$ increases with an increasing $RH_{2m}$. On the other hand, with the same $LCC$ (e.g. $LCC = 40\%$), $RH_{2m}$ decreases with an increase $\overline{w'\theta'}$. Larger $\overline{w'\theta'}$ at high elevation (or low density) region increases the moisture transport from the subcloud layer into the cloud layer. With the same $LCC$ for different air densities, the average relative humidity with higher air density will be greater than that with low air density.

**Appendix A:**
In the main text, the role of environmental descent, $w_e \neq 0$, is explicitly taken into account (cf., Eq. 2). However, this effect is simply neglected (cf., Eq. A1) without any justifications.

The purpose of Appendix A is to demonstrate via a simple model the sole effect of density on the buoyancy flux and its impact on the growth rate

of the CBL depth independent of other factors. The results of $w_s$ for all the LES experiments in this manuscript are identical from the surface to 6 km, which are derived from ERA-interim reanalysis data (Figure B2). As shown in Figure B2(c), from 10:00 LST to 13:00 LST, maximum $w_s \approx 3$ hPa hour$^{-1} \approx 10^{-2}$ m s$^{-1}$. The rate of change of $h$ $dh/dt$ and the entrainment velocity $w_e$ are about one order of magnitude larger than $w_s$; thus $dh/dt$ mainly depends on $w_e$ rather than $w_s$. In order to make the LES simulated profiles as close as possible to the observed profiles from radiosondes (Please refer to Figure B3), we take into account $w_s$ in LES experiments. However, if the effect of $w_s$ is neglected in LES experiments, the main conclusion in this manuscript does not change.

Throughout the Appendix, the prime sign is missing everywhere for indicating the eddy values. The temperature, T, must be replaced by the potential temperature, $\theta$, throughout.

We will add prime sign. T is replaced by $\theta$, but this modification will not affect the conclusion.

Eq. (A9) does not make any sense.

Eq. (A9) is an intermediate step in the derivation.

Eq. (A11) is clearly not a solution of Eq. (A10), thus all the subsequent discussions are irrelevant. In fact, a solution for Eq. (A10) is given by Eq. (5) with $\alpha = 2$, as explicitly stated in Zhu et al (2005). [It is obvious that the authors are citing this paper without reading it. Also note that this solution (5) is earlier derived by Betts (1973, see his Eq. 42). However, the authors simply fail to give such a simple credit.]

In fact, the solution given by Eq.(A11) is equivalent to Eq.(5) with $\alpha = 2$ (i.e., $w_s = 0$), multiplied by $h^{((1+\beta_1)/\beta_1)}$ with an added constant of integration $C$.

**Specific Comments**

Line 88, EDDYPRO: It is a totally non–essential matter for readers what software the authors have used. However, it is crucial to present how turbulent fluxes are actually computed based on which theory, formula, etc.

Thanks for your suggestion. EddyPro Software is used for processing raw eddy covariance (EC) data from the sonic anemometers and gas analyzer to compute atmospheric fluxes of trace gases such as $CO_2$, $H_2O$, $CH_4$, as well as energy, which is widely used in atmospheric sciences, ecology, and related topics. In the manuscript, we pointed out that turbulent fluxes were computed using the eddy covariance technique, and we have provided the website address of EddyPro. We can add a detailed description of theory, formula and method, etc. if that is deemed necessary.

Line 107, LCC: Please spell this out.

We have spelled LCC in line 72.

Sec. 3 (Lines 106–120): Please re–write the text to the point. One has to read it dozen times to understand this rather conjugated text to understand what the authors really want to say. As it stands for now, the remaining part of the main text could be read equally well even by totally removing Sec. 3.

Thanks. We will revise the relevant content, and improve the text, so that the readers will have a clear understanding of the main idea of the text.

Lines 137–146: This paragraph is not linked to any other part of the text. One can simply remove it.

As discussed in section 3 and 4, one of the important variables for a thermal that reaches the lifting condensation level (LCL) is the relative

humidity at the top of the mixed layer (ML) $RH_h$. The paragraph between lines 137–146 is mainly to illustrate the relationship among the height of the mixed layer $h$, $RH_h$, and the relative humidity at the top of the surface layer $RH_0$, so we think it important to leave it in place.

Line 169: Sullivan et al (1998) is just one of many papers quoting an earlier claim with a constant 0.2. This is just an example of arbitrary quotation system that the authors use. The earliest claim of this value that I could dig is Ball (1960). However, the authors must search carefully.

Here we quote Sullivan et al (1998) because this includes a fairly recent summary of extensive research over the years that have reinforced the validity of this value, as well as quantifying its dependency on other parameters, not because it is the earliest claim of this value. We do not find the content that Ball (1960) used this value, please tell us the specific position.

Eq. (5): Indeed this is a valid solution for Eq. (A10) with $\alpha = 2$ when an environmental descent is absent, i.e., $w_e = 0$. However, Zhu et al. (2002) did not derive (cf., Line 169) this expression as a general solution for, say, the system with Eq. (2). They merely suggest it as a phenomenological generalization of the solution with $\alpha = 2$. This is an example of the present authors' system of arbitrary quotations of earlier results out of context.

Based on Eq.(5) of Zhu et al. (2002), $\alpha$ is a subsidence-dependent parameter whose likely maximum range is between 1 and 2. For $w_s = 0$, $\alpha = 2$; and for $dh/dt = 0$ (i.e. $w_e + w_s = 0$), $\alpha = 1$.

Lines 179–181: This premise is hardly justified, which also makes the present manuscript invalid. A work based on such an ill–posed premise should not be published.

We have previously answered this question.

Eq. (7): This is just another example of the present authors' system of arbitrary quotations. This formula is just a curve fit out of an LES, and there is no reason to believe its universality.

Here we compare the parameterization results with LES, not only to describe the differences between them, but also point out that smaller density favors the formation and development of cumulus.

Line 311, "Water vapor is relatively abundant over ECMR in summer": This is a very odd manner to start a final section of a paper considering the role of the low density with a high altitude.

Thanks. The relevant content will be revised. The purpose of the text (Line 311-313) is to illustrate the fact that the relative humidity over ECMR is not less than that over the mid-eastern TP, but high *LCC* occurs mainly over the mid-eastern TP rather than ECMR during summer.

Lines 316–317, "This density effect is demonstrated with a simple mixed–layer model in Appendix A": NO

We have previously answered this question.

Lines 321–322, "Stronger ascending motions": This is nothing to do with a low density

The stronger ascending motions within the thermal are mainly caused by larger $\overline{(w'\theta_v')}_s$ for low density.

Lines 452–454, "a simple microphysics scheme (Grabowski et al 1998) that considers the impact of the relatively low temperature over the TP": NO

Thanks for your suggestion. Of course, there are shortcomings in this microphysics scheme, but the selection of microphysics scheme does not affect the main conclusion in this manuscript.

Line 477, "corrected": Please describe what kind of corrections are made

As described in Appendix B, in order to make sure the results of the LES simulation are close to the observations, we change the original *H* and *LE* results calculated by the eddy covariance method.

Fig. 4: plots are so scattered that I do not think that we can draw any sensible conclusions out of them.

Due to the complexity of the problem itself, the plots are scattered, but the fitting coefficients in the LES sensitivity tests have an obvious trend for varying thermal turbulence and relative humidity. The coefficient of determination is important, which indicates the need for more work in the future.

References

Pepin, N. C., and D. J. Seidel: A global comparison of surface and free-air temperatures at high elevations, *J. Geophys. Res. Atmos.*, 110, D03104, doi:10.1029/2004JD005047, 2005.

Xu, X., Lu, C., Shi, X., and Ding, Y.: Large-scale topography of china: a factor for the seasonal progression of the meiyu rainband?. *J. Geophys. Res. Atmos.*, 115, D02110, doi:10.1029/2009JD012444, 2010.

Wu, G. X., He, B., Duan, A. M., Liu, Y, M., and Yu, W.: Formation and Variation of the Atmospheric Heat Source over the Tibetan Plateau and Its Climate Effects. *Adv. Atmos. Sci.*, 34, 1169–1184, doi: 10.1007/s00376-017-7014-5, 2017.

Wang, Y. J., Xu, X. D., Liu, H. Z., Li, Y. Q., Li, Y. H., Ze, Z. Y., Gao, X. Q., Ma, Y. M., Sun, J. H., Lenschow, D. H., Zhong, S. Y., Zhou, M. Y., Bian, X. D., and Zhao. P.: Analysis of land surface parameters and turbulence characteristics over the Tibetan Plateau and surrounding region, *J. Geophys. Res. Atmos.*, 121, 9540-9560, doi:

10.1002/2016JD025401, 2016.

[Figure]

Fig. 1 (a) The relationships among monthly means of $LCC$, $\rho_{2m}$ and $RH_{2m}$ observed by the AWS in summer. The samples are divided into two groups: $RH_{2m} > 72\%$ (red dots) and $RH_{2m} < 72\%$ (blue dots). Region A and region B generally correspond to $RH_{2m}$ greater than and less than 72%, respectively. The histogram shows an approximate relationship between $\rho_{2m}$ and surface elevation above sea level $z$ at the bottom of

Figure 1 (a).

[Figure]

Fig. S1 The relationship between elevation and (a) $H$, (b) $T_g$-$T_a$, (c) $\overline{w'\theta'}$ from ERA-interim data from about 9:00 LST to 15:00 LST (3:00 UTC to 9:00 UTC) in East Asia in summer. The solid circles denote median values. The error bars represent 25% and 75% probability values respectively.

[Figure]

Fig. S2 The relationships among monthly means of *LCC*, $\overline{w'\theta'}$ and $RH_{2m}$ from ERA-interim data from about 9:00 LST to 15:00 LST (3:00 UTC to 9:00 UTC) in summer.

---

## Author Comment (AC3) · 28 Aug 2019

Responses to the interactive comments on "The effect of low density over the "roof of the world" Tibetan Plateau on the triggering of convection" by Referee#3 (Authors)

Thanks for your comments on the manuscript.

At first, we illustrate the main purpose and innovation of this study. As mentioned by referee #2, this manuscript presents an interesting approach to looking at variations in convective activity over large orography, based on the corresponding variations in air density at the surface. The observation data show that there is stronger thermal turbulence and higher frequency of low cloud formation over the eastern and central TP which provides a basis for our study of whether there is a relationship among the formation and evolution of frequent convective clouds, low air density, and turbulence generation over the TP. We consider this from two aspects: climate statistics and large-eddy simulation (LES).

Specific comments:

l 79 : "interrelation between turbulence and convective motion" : what do you mean here ? This is too vague, and should be better introduced. What has been done before and what is different here.

Thanks. We have rephrased the sentences. Our intention is to emphasize the effect of thermal turbulence on cumulus convection.

l70-71 : very vague. Delete or rephrase. It is not very clear from the introduction what are the "scientific issues"

Please see the first paragraph.

L114-116 : Why is that? Any idea?

The above statistical results from climate data are closely related to the scientific issue discussed and studied in this manuscript. The above phenomenon is explored with the LES experiments, and our main points are as follows: Larger $\overline{w'\theta'}$ at high elevation

(or low density) regions increase the moisture transport from the subcloud layer into the cloud layer, which favors cloud formation. Therefore, with the same relative humidity as a low elevation region, more low cloud exists over the TP.

Fig1a : Are there blue dots under the red ones? It looks like it. Consider using transparency to make the figure more readable and to fully demonstrate your point.

Thanks for your suggestion. We have changed the transparency of the dots, and first plot the dots with low $RH_{2m}$ ($RH_{2m} < 72\%$) which makes figure 1 (a) easier to interpret.

L 154: define the cloud cores here. How do you get them from the model?

The cloud core is defined as the grids with liquid (or ice) water content greater than zero and the virtual potential temperature $\theta_v$ greater than average potential virtual temperature $\overline{\theta_v}$ at the same height. The results of the cloud core are derived from the three-dimensional liquid water content, ice water content, and $\theta_v$ output fields.

L157-158 : justify and explain physical meaning of this assumption.

L158-159 : confusing: : :. Is your model general or only applying to your specific study. Clarification needed. And the sentence must be rephrased.

For the convective boundary layer case of this study, as shown in Figure 2 (b), $a_{cc} = 0\%$ before LST 12:00, thus $M = 0$ m s$^{-1}$ according to Eq. (3). The cloud core has just appeared at about 13:00 LST when $dh/dt$ is on the order of $10^{-1}$ m s$^{-1}$, while $w_{cc}$ is on the order of 1 m s$^{-1}$. Therefore, when $a_{cc}<1\%$, $dh/dt$ is at least about one order of magnitude larger than $M$ according to Eq. (3) so we can ignore $M$ when we use Eq. (2) to calculate $w_e$. $M$ is equal in magnitude to $w_e$ in the developmental stage of cumuli due to larger $a_{cc}$ (after about 15:00 LST), thus $M$ cannot be ignored in that case.

The Dutch Atmospheric Large-Eddy Simulation (DALES) model is an effective tool for studying the shallow cumulus clouds that can form on top of dry rising thermals in the subcloud layer and thus is suitable for studying the scientific issues in this manuscript. This manuscript mainly discusses the growth rate of the convective

boundary layer and the effect of thermal turbulence on the formation and evolution of cumulus in daytime, so we mainly focus on the effect of surface heating especially for varying air density.

As shown in Figure B2(c), from 10:00 LST to 13:00 LST, the maximum $w_s \approx 3$ hPa hour$^{-1} \approx 10^{-2}$ m s$^{-1}$. $dh/dt$ and $w_e$ are about one order of magnitude larger than $w_s$, thus $dh/dt$ mainly depends on $w_e$ rather than $w_s$. Here we use eq. (2) to calculate $w_e$.

Subsection 4.1: Equations are used out of context without discussing the underlying assumptions and the situation to which they apply. This should be done everytime it is releveant. It is also unclear what is the goal here. Is it to build equation (6)? Under what assumptions and for what purpose?

In subsection 4.1, the main point we want to illustrate is, compared to the high $\rho$ case, the growth rate of $h$ increases faster for the low $\rho$ case due to larger $\overline{(w'\theta'_v)}_s$. According to the Eq. (1), with the same $RH_0$, larger $RH_h$ and more favorable conditions for saturation occur for small $\rho$ compared to large $\rho$. (Sorry, there is a mistake in line 194-195, the four LES experiments should be (CON, $1.2\rho_{CON}$, $1.4\rho_{CON}$, $1.7\rho_{CON}$) rather than (CON, $1.4\rho_{CON}RH0.05$, $1.4\rho_{CON}RH0.15$, $1.4\rho_{CON}RH0.3$)). Whether or not the penetrative convection can form cloud is related not only to the strength of ascending motion within the thermal but also to relative humidity ( reflected by $RH_h$). The goal of introducing Eq. (1) is to show the relationship between the relative humidity at the top of the surface layer $RH_0$ and the relative humidity at the top of the mixed layer (ML) $RH_h$. We have stated the assumptions of Eq. (1) in line 137-138. In most cases, these assumptions are reasonable and within the CBL, the water vapor mixture ratio $q_T$ is invariant (or slightly decreases) with increasing height and adiabatic temperature lapse rate $\partial T/\partial z = -\gamma_d$.

L199: There is a lack of discussions of the BL structure in the observations and the model, and a figure like Fig B2 should be in the main part of the paper, and compared

with the model results. Make it clear that penetrative convection means here dry convection making it through the inversion and then possibly forming a cloud. Or remove this sentence as more explanations follow it anyway.

Thanks. The related content has been revised, and we have added discussions of the BL structure in the observations and the model results.

L213-214: unclear, be more explicit.

The local CBL height $h_{local}$ for a specific point (x, y) is defined as the lowest level for which the local virtual potential temperature gradient is larger than 2 K km$^{-1}$, and the penetration depth $d_t$ for this specific point (x, y) is defined as the difference between $h_{local}$ and $h$.

Section 4.3: this section must be better organised and clarified. More hindsight is needed to avoid only going back and forward from a specific sub-result to one of several cherry-picked models from the literature. What is the reference you are comparing to?

Where are the results from the different schemes? If you decide to not show them this must be made very clear.

Thanks. We have revised section 4.3 to further clarify our point and better explain the climate data shown in Figure 1.

L 454-455 : This surely cannot be the resolution, but must be the domain size. Specify the corresponding resolution and justify the choice of the model top.

Thanks for your suggestion. The descriptions of domain size and resolution are inaccurate. The LES were performed on a numerical domain of 128 x 128 x 150 grid points. The horizontal and vertical resolutions are 50 m and 40 m, respectively. In our LES experiments, the maximum cloud top height is about 4.5 km in the afternoon, thus we think 6 km is a reasonable value for the model top.

Technical :

Only a few comments here as it is not relevant at this state, again not exhaustive at all

(cf. My general comments) l133 : "We analyse in detail the results of control experiment ..." : be more precise by adding "In this section ..." l137: it's "water vapor mixing ratio" Eq. (2) : partial derivatives should be used here (no Large scale BL advection is assumed as far as I understood) L 456 : replace by 14 hours, this is more readable.

Thanks. We have made the suggested changes. As shown in Figure B2 (d) and (e), we have taken into account large-scale advection in the LES experiments. Of course, here the height-based vertical coordinate may be more appropriate due to different pressure at the same height for different LES experiments.

[Figure]

Fig. 1 (a) The relationships among monthly means of $LCC$, $\rho_{2m}$ and $RH_{2m}$ observed by the AWS in summer. The samples are divided into two groups: $RH_{2m} > 72\%$ (red dots) and $RH_{2m} < 72\%$ (blue dots). Region A and region B generally correspond to $RH_{2m}$ greater than and less than 72%, respectively. The histogram shows an approximate relationship between $\rho_{2m}$ and surface elevation above sea level $z$ at the bottom of Figure 1 (a).

---

## Author Comment (AC4) · 28 Aug 2019

Thank you very much for the comments on the manuscript. Here is our response.

**General comments**

1. Although the observation and reanalysis data used are briefly described in Section2, there appears to be no section describing the LES model in the main paper. Without knowing which model is used, its domain size and resolution and major paramterisations etc., it is impossible to understand the likely impact of model errors in the results presented here. While it's fine to move further detail to an appendix, the basic information should be presented in the main text prior to the presentation of any results.

   We have added the basic descriptive information (domain size, resolution and major parameterization scheme) about the LES model in the main paper.

2. In various places, the variation of behaviour with air density is considered, but it is often not made clear to what extent this means the fixed variation of density due to the orography, or the synoptic variations which may occur at any given location.

   Thanks. We agree that synoptic variations are usually accompanied by a change of pressure and temperature, and thus density will change at any given location. However, because the density difference between TP and low elevation regions is mainly due to the height above sea level rather than synoptic variations, we do not discuss this issue in the manuscript. For example, the surface pressures in the four LES experiments (CON, $1.2\rho_{CON}$, $1.4\rho_{CON}$, $1.7\rho_{CON}$) are about 580 hPa, 695 hPa, 810 hPa and 985 hPa, respectively. Their corresponding elevations are about 4.5 km, 3.0 km, 2.0 km and 0.2 km, respectively. At any given location, the variations

of surface pressure caused by synoptic variations are usually confined to $\pm 20$ hPa. However, in subsequent studies we may consider adding analysis and discussion of the effects of synoptic pressure and temperature variations on LES results.

3. There is considerable discussion of relationships between density $\rho$ and CBL height $h$. However, it is not obvious that comparing a geometric height/thickness measure across large variations in density is appropriate. Consideration should be given to how the relationships would look with a mass-based measure of thickness (corresponding to a pressure-based rather than height-based vertical coordinate). The same applies to the question of variations in vertical velocity with density – these relationships may look very different between geometric vertical velocity and pressure velocity).

Thanks for your suggestion. An excellent question. We replotted Figure 2(a) and Figure 3 with the vertical coordinate changed to a pressure-based coordinate, in Figures N2(a) and N3. $h_p$ is the mixed-layer thickness expressed in units of hPa in Figure N2(a). There are no significant differences in $h_p$ for the four LES experiments (CON, $1.2\rho_{CON}$, $1.4\rho_{CON}$, $1.7\rho_{CON}$) before 12:30 LST. The growth rate $dh_p/dt$ slows down for lower density experiments after 12:30 LST. For large $\rho$, $h_p$ is significantly larger than for small $\rho$. The main reason for this is that the rapid growth rate of $h$ for small $\rho$ makes up for the effect of $\rho$ in the morning. Compared to the high $\rho$ case, the thicker $h$ results in an obvious decrease of $w_e$ and $dh/dt$ for small $\rho$ in the afternoon. The relationships between geometric vertical velocity $w$ and pressure velocity $\omega$ can be written as: $\omega = -\rho g w$. Compared to the low $\rho$ case, the large $\rho$ makes up for the effect of the weaker local $w$ (Figure N3 (d) X $\approx$ 3.0 km), thus there are no significant differences in pressure velocity $\omega$, and in the penetration depth $d_{th}$ expressed in units of hPa as shown in Figure N3 (c) and (d).

**Specific comments**

p.4, line 119 Kelvin are not degrees, i.e. the unit is K, not °K.

Thanks, it has been revised.

p.3, lines 75–76 Are "very small" horizontal scales of "tens of metres" adequately resolved by the LES configuration used in the study?

p.454, lines 454–455 A resolution on the order of 6km is not an LES model, but in the realm of the highest-resolution global NWP models, or "cloud-system resolving models". Or should this read "A domain size of 6.4km x 6.4km x 6.0km. . . "? This would make the horizontal and vertical resolution 250m and 40m respectively, which seems more reasonable, but still unable to resolve well the "tens of metres" scale referred to on lines 75–76.

Thanks for your suggestion. The descriptions of domain size and resolution were inaccurate. The LES are performed on a numerical domain of 128 x 128 x 150 grid points. The horizontal and vertical resolutions are 50 m and 40 m, respectively.

Figures 2, 4 The panels should be formatted so that legends do not obscure the actual data points.

Thanks, it has been revised.

Figure 3 The (a), (b), (c), (d) labels in white can barely be read against the patterns in the actual plots. These labels should probably be moved outside the plot area.

Thanks, it has been revised.

p.17, lines 520–527 The original datasets used are stated to be available "upon request", rather than being deposited in a readily-accessible archive. I would draw attention to this, but leave it to the editor's discretion whether this is sufficient to meet the journal's data policy without further justification.

Thanks for your suggestion. Due to the huge volume of data, here we did not upload the reanalysis data, which is available at https://apps.ecmwf.int/datasets/data/interim-full-mnth/levtype=sfc/ and https://apps.ecmwf.int/datasets/data/interim-full-daily/levtype=sfc/. We are certainly

willing to upload all other data used in this manuscript (include the surface and radiosonde observation and LES results).

[Figure]

Fig. N2 The time variations of (a) mixed layer thickness $h_p$ expressed in units of hPa

for the four LES experiments (CON, $1.2\rho_{CON}$, $1.4\rho_{CON}$ , $1.7\rho_{CON}$)

[Figure]

[Figure]

Fig. N3 Same as Figure 3 (c) and (d), but with a pressure-based vertical coordinate.
The X-axis wind speeds are not plotted in the Figure.